# 2-Nitroimidazoles induce mitochondrial stress and ferroptosis in glioma stem cells residing in a hypoxic niche

Naoyoshi Koike [1,2], Ryuichi Kota[1,2], Yoshiko Naito[3], Noriyo Hayakawa[3], Tomomi Matsuura[3], Takako Hishiki[3,4], Nobuyuki Onishi[1], Junichi Fukada[2], Makoto Suematsu [4], Naoyuki Shigematsu[2], Hideyuki Saya[1] & Oltea Sampetrean [1✉]

Under hypoxic conditions, nitroimidazoles can replace oxygen as electron acceptors, thereby enhancing the effects of radiation on malignant cells. These compounds also accumulate in hypoxic cells, where they can act as cytotoxins or imaging agents. However, whether these effects apply to cancer stem cells has not been sufficiently explored. Here we show that the 2-nitroimidazole doranidazole potentiates radiation-induced DNA damage in hypoxic glioma stem cells (GSCs) and confers a significant survival benefit in mice harboring GSC-derived tumors in radiotherapy settings. Furthermore, doranidazole and misonidazole, but not metronidazole, manifested radiation-independent cytotoxicity for hypoxic GSCs that was mediated by ferroptosis induced partially through blockade of mitochondrial complexes I and II and resultant metabolic alterations in oxidative stress responses. Doranidazole also limited the growth of GSC-derived subcutaneous tumors and that of tumors in orthotopic brain slices. Our results thus reveal the theranostic potential of 2-nitroimidazoles as ferroptosis inducers that enable targeting GSCs in their hypoxic niche.

[1] Division of Gene Regulation, Institute for Advanced Medical Research, Keio University School of Medicine, Tokyo, Japan. [2] Department of Radiology, Keio University School of Medicine, Tokyo, Japan. [3] Clinical and Translational Research Center, Keio University School of Medicine, Tokyo, Japan. [4] Department of Biochemistry, Keio University School of Medicine, Tokyo, Japan. ✉email: oltea@a6.keio.jp

ntratumoral regions exposed to low contents of molecular oxygen ($O_2$) harbor therapy-resistant cancer cells and are a key therapeutic target[1]. Although no single drug is able to eradicate all hypoxic malignant cells, two major pharmacological approaches to their specific targeting have been developed: inhibition of proteins that function in the cellular response to hypoxia, and the administration of hypoxia-selective drugs[1,2].

Among compounds that are able to exert anticancer effects in the absence of oxygen, nitroimidazoles are considered promising theranostic agents[3] for several reasons. The electrophilic properties of nitroimidazoles allow them to oxidize the radiation-induced DNA radicals[4–6], thereby enhancing the effects of radiation. Furthermore, on entering cells by diffusion, nitroimidazoles undergo one-electron reduction of the nitro group. In the presence of $O_2$, the reduced metabolites of these compounds are immediately reoxidized, resulting in redox cycling of the drugs. However, in the absence of oxygen, the reduced derivatives are stabilized and accumulate within the cell, where they can undergo further serial reduction[7,8]. This characteristic has allowed several 2-nitroimidazoles to serve as exogenous markers of hypoxia. Pimonidazole and etanidazole pentafluoride (EF5) have thus been widely adopted in preclinical studies[9], whereas misonidazole is applied clinically in the form of [18F]fluoromisonidazole as a hypoxia tracer for positron emission tomography[10]. In addition, nitroreduction of 2-nitroimidazoles has been reported to result in the production and accumulation of toxic metabolites as well as in the consumption of reducing equivalents under anaerobic conditions, thus, leading to direct, radiation-independent cytotoxic effects on hypoxic cells[8].

The biological implications of the bioreductive metabolism of nitroimidazoles in cancer have not been fully explored, however, which has limited their therapeutic application. The 5-nitroimidazole nimorazole has been shown to significantly improve the efficacy of radiotherapeutic management in cancer patients[11]. In contrast, although they possess a higher electron affinity, 2-nitroimidazoles showed only a moderate effect in this regard[12,13], with this inadequacy having been attributed to dose-limiting neurotoxic side effects[4,13]. However, the recent finding that cancer cells with stem-like properties often reside in hypoxic lesions[14] has suggested that the limited therapeutic effect of 2-nitroimidazoles might also reflect a resistance of cancer stem cells to this class of compounds.

Doranidazole, or 1-(1′,3′,4′-trihydroxy-2′-butoxy)-methyl-2-nitroimidazole (PR-350), was designed to reduce the neurotoxicity of 2-nitroimidazoles by limiting blood–brain barrier permeability[15–17]. It has shown radiosensitizing effects both in vitro and in vivo in preclinical studies[18,19], and its administration before radiotherapy contributed to long-term survival in patients with unresectable pancreatic or locally advanced non-small cell lung cancer[20,21]. It has remained unclear, however, whether doranidazole is able to serve as a radiosensitizer for cancer stem cells and whether it exerts yet unidentified radiation-independent cytotoxic effects on hypoxic cells.

We have now examined these issues with the use of a mouse model of glioblastoma (GBM), a highly hypoxic tumor type[22]. We found that doranidazole enhanced radiation-induced DNA damage in hypoxic glioma stem cells (GSCs) and conferred a survival benefit in mice harboring GSC-derived tumors in the radiotherapy setting. Furthermore, doranidazole and misonidazole, but not metronidazole, also induced mitochondrial dysfunction and ROS accumulation in GSCs, resulting in the induction of ferroptosis and a reduction of the hypoxic GSC niche, shedding light on possible theranostic applications of these compounds.

## Results

### Radiosensitizing effect of doranidazole on GSCs.

To investigate radiosensitization by and hypoxic cell-specific toxicity of nitroimidazoles, we adopted our previously established syngeneic mouse GBM model[23]. Glioma-initiating cells (GICs) were established by transduction of *Ink4a/Arf*-null neural stem/progenitor cells (NSCs) with a vector for the oncoprotein H-Ras[V12] (but without a fluorescent marker). The GIC-H cells obtained after selection with hygromycin expressed the stem cell markers Nestin and SOX2, underwent glial differentiation on exposure to fetal bovine serum (FBS), and formed heterogeneous and aggressive tumors resembling human GBMs on orthotopic implantation (Supplementary Fig. 1a–d). Cells with stem-like properties were isolated from late-stage intracranial tumors that had interacted with the syngeneic environment and formed hypoxic regions. The resulting GSC-H cells showed increased tumor formation potential and radioresistance compared with GIC-H cells (Supplementary Fig. 1c–e), and they were adopted for subsequent experiments.

We first examined the effect of doranidazole on the radiosensitivity of GSC-H cells. Mice were exposed to 15 Gy of ionizing radiation, with or without prior intraperitoneal (i.p.) administration of doranidazole (200 mg/kg), at 10 days after orthotopic implantation of GSC-H cells. Doranidazole administration combined with radiation resulted in a significant prolongation of survival compared with radiation alone (Fig. 1a). In contrast, a single dose of doranidazole had no effect on survival in the absence of radiation (Fig. 1a). Cultures of GSC-H cells maintained in the presence of <0.1% $O_2$ showed a significant increase in the number of foci positive for histone γH2AX, a marker of DNA double-strand breaks, on exposure to radiation in the presence of doranidazole compared with irradiation alone, whereas doranidazole had no such effect under normoxic conditions (Fig. 1b, c) and had no effect in the absence of radiation (Supplementary Fig. 1f). In a colony formation assay, doranidazole had a significant radiosensitizing effect on GSC-H cells that was more pronounced under severely hypoxic (<0.1% $O_2$) than under normoxic conditions (Fig. 1d, e). A 3-h exposure to doranidazole alone tended to reduce the plating efficiency of GSC-H cells in the presence of either 20% or <0.1% $O_2$ (Fig. 1f, g), but this effect did not achieve statistical significance.

Together, these results showed that doranidazole mediates radiosensitization of both GSCs and GSC-based tumors.

### Doranidazole induces GSC death under hypoxic conditions.

In addition to regions of severe hypoxia or anoxia (<0.1% $O_2$; binding of hypoxia-marker EF5, 30–100%)[24], which are primary targets of radiosensitization, GBMs contain regions with $O_2$ levels corresponding to 0.5–2.5% (EF5 binding, 3–10%)[24], usually considered as "modest" or "mild" hypoxia[24,25]. Given that hypoxic conditions of 1–2% $O_2$ increase the stem cell fraction and cell proliferation in GBMs[25,26], we next asked whether 2-nitroimidazoles possess additional radiation-independent cytotoxic effects on GSCs under mild hypoxia.

Flow cytometric analysis of the cell cycle showed that the proportion of GSC-H cells with a DNA content of between 2N and 4N (corresponding to S phase) declined, whereas that of those with a content of 4N increased, after exposure to doranidazole (Supplementary Fig. 2a, b). The effects of doranidazole on the cell cycle were similar under both normoxic (20% $O_2$) and hypoxic (1% $O_2$) conditions, with the exception that the drug also induced an increase in the proportion of cells with a DNA content of <2N (representing dead cells) under the hypoxic condition. Equimolar concentrations of metronidazole and

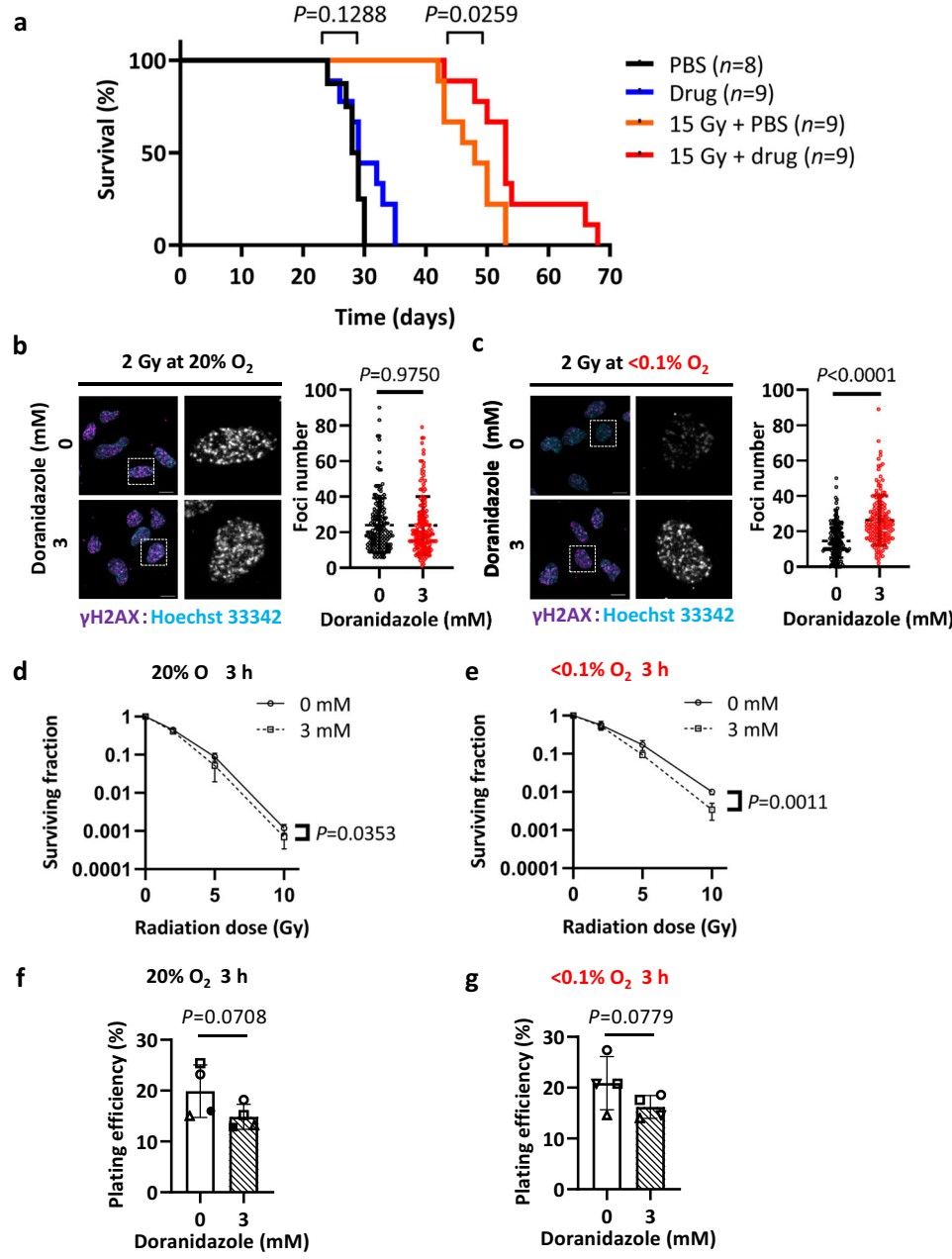

**Fig. 1 Radiosensitizing effect of doranidazole on GSCs. a** Survival curves for the indicated numbers (n) of mice exposed to 0 or 15 Gy of radiation 10 days after orthotopic implantation of 1000 GSC-H cells, with or without administration of doranidazole (200 mg/kg, i.p.) 30 min before irradiation. PBS phosphate-buffered saline. Representative experiment, n = 2 independent experiments performed. GSC-H cells cultured under normoxic (**b**) or hypoxic (**c**) conditions were incubated for 30 min in the absence or presence of 3 mM doranidazole, exposed to 2 Gy of radiation, and then subjected to immunofluorescence staining of γH2AX at 30 min after irradiation. Nuclei were stained with Hoechst 33342. The boxed regions of the left images are shown at higher magnification in the right images. Scale bars, 10 μm. The number of γH2AX foci per nucleus was determined for all nuclei from n = 3 independent experiments, including a total of >150 nuclei per group. GSC-H cells were incubated for 3 h in the absence or presence of 3 mM doranidazole under normoxic **d**, **f** or hypoxic **e**, **g** conditions, exposed to the indicated doses of radiation **d**, **e**, and then subjected to a colony formation assay for 10 days under normoxic conditions. The surviving fraction **d**, **e** and plating efficiency **f**, **g** determined as the mean ± s.d. from n = 3 **d**, **e** or n = 4 **f**, **g** independent experiments are shown. Statistical analysis was performed with the log-rank test **a**; the unpaired two-tailed Student's t-test **b**, **c**; two-way ANOVA followed by Sidak's post hoc test **d**, **e**, with the P values being for comparison of plus or minus doranidazole at 10 Gy; or the paired two-tailed Student's t-test **f**, **g**.

misonidazole did not affect cell cycle distribution (Supplementary Fig. 2b).

To evaluate the cell cycle profile in more detail, we generated a bicistronic vector encoding mCherry-hCdt1 and EGFP-hGem fusion proteins (Supplementary Fig. 2c) that was based on the FUCCI (fluorescent ubiquitination-based cell cycle indicator) system[27]. The vector was validated in mouse embryonic

fibroblasts to confirm that cells in $G_1$ phase express mCherry, those in $S$–$G_2$–$M$ phases express EGFP (enhanced green fluorescent protein), and contact inhibition induces an accumulation of mCherry-positive cells (Supplementary Fig. 2d). The construct was then introduced into GSC-H cells, with the resulting cells being designated GSC-F. Time-lapse imaging of GSC-F cells exposed to doranidazole under the normoxic

condition revealed a gradual decline in the number of EGFP-positive ($S$–$G_2$–M) cells and an accumulation of mCherry-positive ($G_0$–$G_1$) cells (Supplementary Fig. 2e), suggesting that the doranidazole-induced increase in the 4N population of GSC-H cells detected by flow cytometry (Supplementary Fig. 2a) reflected the appearance of cells that had undergone mitotic slippage and entered the subsequent $G_1$ phase.

To evaluate cytotoxicity, we quantified the binding of propidium iodide (PI) to DNA in unfixed GSC-H cells. Flow cytometry revealed that doranidazole and misonidazole, but not metronidazole, each induced a significant increase in the proportion of dead cells after drug treatment for 72 h in the presence of 1% $O_2$ (Fig. 2a, b). Of note, the induction of cell death by doranidazole was apparent earlier and was more pronounced in the presence of severe hypoxia (<0.1% $O_2$) (Fig. 2a, Supplementary Fig. 2f), consistent with less redox cycling of the drug. Serum-treated NSCs, which differentiate along the astrocytic line (Supplementary Fig. 1b) underwent significant cell death after exposure to doranidazole in the presence of 0.1% $O_2$, but not in the presence of 20% or 1% $O_2$ (Supplementary Fig. 2g).

We next examined cell death in a model that more closely mimics tumors with regard to the presence of regions with different oxygen tensions. GSC-based spheres possess not only a three-dimensional structure but also a hypoxic core, as revealed by EF5 staining (Fig. 2c). Exposure of GSC-H spheres to doranidazole for 24 h induced cell death (as revealed by PI staining) in the central region of the spheres in a concentration-dependent manner (Fig. 2d). Exposure to 3 mM doranidazole for a period as short as 6 h also induced marked cell death in the central region of the spheres (Supplementary Fig. 2h). The 2-nitroimidazoles misonidazole and etanidazole, but not the 5-nitroimidazoles metronidazole or nimorazole, also manifested a similar effect at equimolar concentrations (Fig. 2e, f).

We next examined the response to doranidazole in two human glioma cell lines. Under monolayer culture conditions, in which cells are uniformly exposed to the same $O_2$ level, treatment with 3 mM doranidazole for 24 h induced significant cell death in both cell lines in the presence of <0.1% $O_2$ (Supplementary Fig. 2i, j). In sphere culture, the GBM cell line U251 formed spheres with an EF5-positive core (Fig. 2g) and showed significant cell death on exposure to doranidazole for 24 h (Fig. 2h). In contrast, spheres formed by the astrocytoma cell line Becker had a necrotic core with a proliferating outer layer, but with minimal EF5-positive hypoxic cells, and were not sensitive to doranidazole (Fig. 2g, h). Together, these results suggested that doranidazole and misonidazole exert pronounced cytotoxic effects on hypoxic GSCs.

**Ferroptosis mediates doranidazole-induced hypoxic GSC death**. To characterize the mechanism of GSC death induced by doranidazole, we first examined the effects of inhibitors of necroptosis, ferroptosis, and apoptosis. Doranidazole toxicity in GSC-H spheres was attenuated markedly by the ferroptosis inhibitor ferrostatin-1 and to a lesser extent by the necroptosis inhibitor necrostatin-1, whereas the apoptosis inhibitor Z-VAD-FMK had no significant effect (Fig. 3a and Supplementary Fig. 3a). The iron-chelator deferoxamine also inhibited doranidazole-induced cell death in a concentration-dependent manner (Fig. 3b). These results thus suggested that doranidazole-induced GSC death is mediated, at least in part, by ferroptosis.

An increase in the intracellular amounts of ROS is one trigger of ferroptosis[28]. Indeed, gene ontology analysis of gene set enrichment analysis pathways for doranidazole-induced changes in the transcriptome of GSC-H cells revealed that "oxidation reduction process" and "response to oxidative stress" were the most upregulated biological processes common to both normoxic and mildly hypoxic (1% $O_2$) conditions (Supplementary Fig. 3b–d). In addition to genes encoding antioxidant enzymes such as NAD(P)H-dependent quinone oxidoreductase 1 (*Nqo1*) and catalase (*Cat*), genes in the "oxidation reduction process" set whose expression was upregulated by doranidazole included those for enzymes related to the reduction of nitroimidazoles—*Txnrd1*[29], which encodes thioredoxin reductase 1—or to ferroptosis, including *Steap3* and *Hmox1*, which encode a metalloreductase[30] and heme oxygenase 1[31], respectively (Supplementary Fig. 3e). Consistent with the doranidazole-induced increase in the expression of genes related to "response to oxidative stress," the antioxidants 10-(6′-plastoquinonyl) decyltriphenylphosphonium (SKQ1)[32,33] and *N*-acetyl cysteine (NAC) partially inhibited doranidazole-induced GSC-H death (Fig. 3c and Supplementary Fig. 3f).

Given that SKQ1 targets mitochondria and that mitochondrial ROS play a context-dependent role in ferroptosis[28], we examined the effect of nitroimidazoles on mitochondrial ROS production in GSC-H cells. Staining of mitochondria in these cells with MitoTracker Green revealed that doranidazole or misonidazole, but not metronidazole, induced a change in mitochondrial morphology from tubular to smaller and rounder (Supplementary Fig. 3g). Staining with MitoSOX Red also showed that doranidazole or misonidazole, but not metronidazole, induced ROS accumulation in mitochondria, with this effect being more pronounced under hypoxic conditions (Fig. 3d, e and Supplementary Fig. 3h). In addition, this effect of doranidazole tended to be inhibited by SKQ1, although not to a statistically significant extent (Fig. 3e). Furthermore, treatment with doranidazole and SKQ1 had similar effects on lipid peroxidation as detected with the fluorescent probe BODIPY C11 (Fig. 3f, g).

**Doranidazole attenuates mitochondrial complex activity in GSCs**. Mitochondrial dysfunction—in particular, dysfunction of mitochondrial complex I and an increase in the NADH/NAD$^+$ ratio—has been found to be a major cause of elevated mitochondrial ROS levels[34]. Assessment of mitochondrial function by extracellular flux analysis showed that the basal oxygen consumption rate (OCR) of GSC-H cells was reduced by incubation with doranidazole under normoxic (20% $O_2$) or hypoxic (1% $O_2$) conditions for 12 h (Fig. 4a, b). Uncoupling of oxidative phosphorylation with carbonylcyanide-*p*-trifluoromethoxyphenylhydrazone (FCCP) revealed that mitochondrial spare respiratory capacity of GSC-H cells was also reduced by incubation with doranidazole (Fig. 4a, b). Similar effects were apparent with misonidazole, but not with metronidazole (Supplementary Fig. 4a). Permeabilization of the plasma membrane allows complex specific substrates to access mitochondria. Mitochondrial complex I-dependent respiration, measured after the addition of pyruvate, was reduced by incubation with doranidazole for 12 h (Fig. 4c), as was mitochondrial complex II-dependent respiration, measured after the addition of succinate (Fig. 4c, d). Mitochondrial complex III- and complex IV-dependent respiration, measured after the addition of duroquinol and *N*,*N*,*N*′,*N*′-tetramethyl-*p*-phenylenediamine (TMPD), respectively, were not substantially affected by doranidazole (Fig. 4c, d).

To investigate the molecular basis of the dysfunction of the mitochondrial respiratory chain induced by doranidazole, we examined protein expression for the complexes by immunoblot analysis. Doranidazole-induced downregulation of mitochondrial complex I and II proteins in a concentration-dependent manner (Fig. 4e), although similar downregulation of the corresponding mRNAs was not apparent (Supplementary Fig. 4b, c). Furthermore, the proteasome inhibitor MG132 did not affect the depletion of mitochondrial complex proteins induced by doranidazole (Supplementary Fig. 4d), suggesting that this depletion was not due to proteolysis. Misonidazole, but not

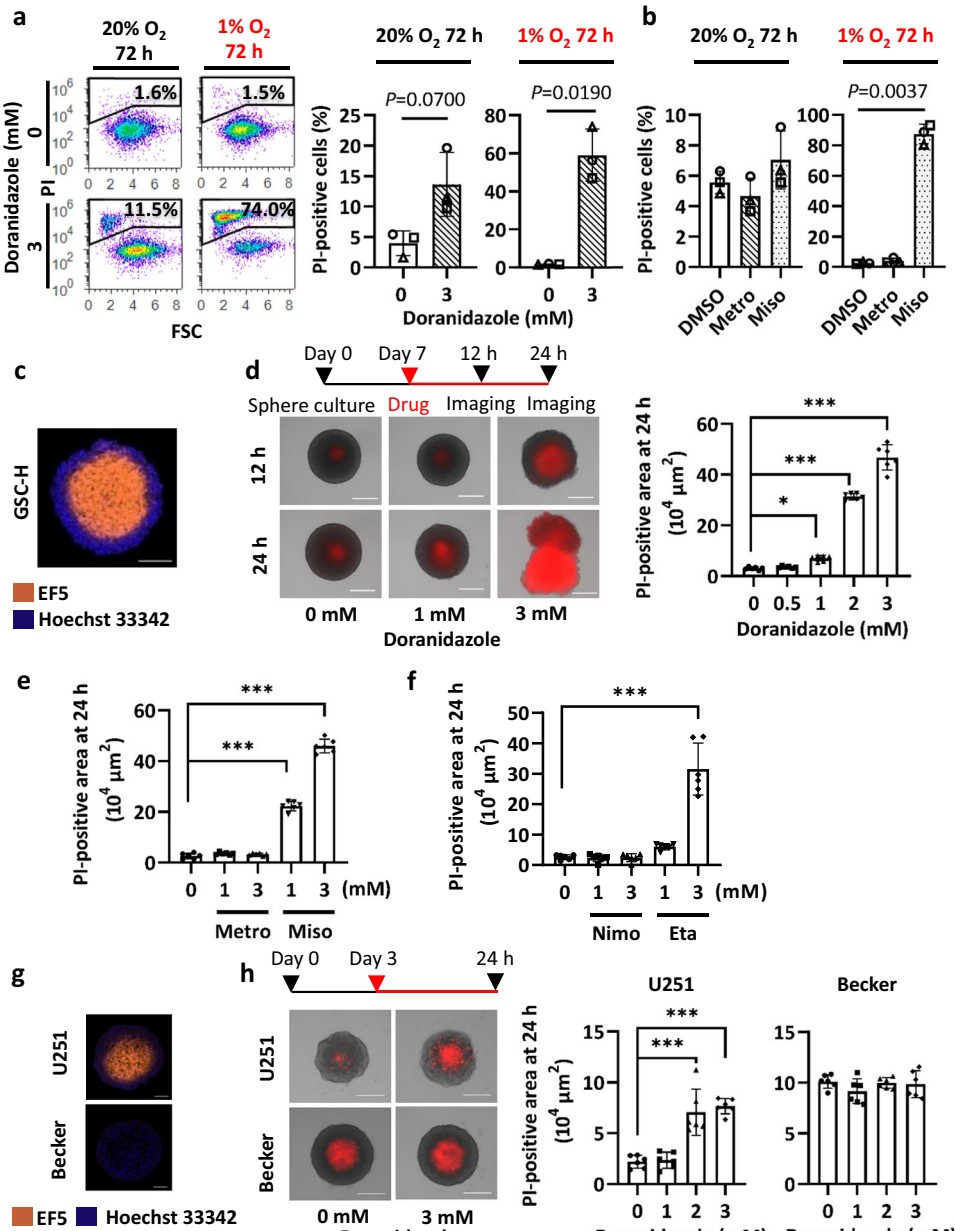

**Fig. 2 Cytotoxic effect of doranidazole on GSCs. a** Flow cytometric analysis of cell death for GSC-H cells exposed to 0 or 3 mM doranidazole under normoxic or hypoxic conditions for 3 days. Representative profiles and quantification of PI-positive cells are shown. FSC forward scatter. **b** Flow cytometric analysis of cell death for GSC-H cells exposed to 3 mM metronidazole (Metro), 3 mM misonidazole (Miso), or dimethyl sulfoxide (DMSO) vehicle under normoxic or hypoxic conditions for 3 days. **c** Immunofluorescence staining for EF5 (marker of hypoxic cells) in a sphere formed by GSC-H cells. Nuclear staining, Hoechst 33342. **d** GSC-H spheres formed over 7 days were exposed to the indicated concentrations of doranidazole for 12 or 24 h, after which dead cells were identified based on PI uptake. Representative images and quantification of the PI-positive sphere area at 24 h are shown. **e** Evaluation of cell death based on PI uptake for GSC-H spheres incubated for 24 h with the indicated drugs or DMSO vehicle. Quantification of the PI-positive sphere area is shown. **f** Evaluation of cell death based on PI uptake for GSC-H spheres incubated for 24 h with the indicated drugs or DMSO vehicle. Quantification of the PI-positive sphere area is shown. **g** Immunofluorescence staining for EF5 incorporation in spheres formed by U251 and Becker cells over 2 days. Nuclear staining, Hoechst 33342. **h** Evaluation of cell death based on PI uptake for U251 and Becker spheres incubated for 24 h with the indicated concentrations of doranidazole. Representative images and quantification of the PI-positive sphere area are shown. Quantitative data for (**a**, **b**) are means ± s.d. for $n = 3$ independent experiments and were analyzed with the paired two-tailed Student's $t$-test (**a**) or one-way ANOVA followed by Dunnett's post hoc test **b**. Data for **d**–**f**, **h** are means ± s.d. of six biologically distinct replicates in a representative experiment ($n = 3$ independent experiments performed) and were analyzed by one-way ANOVA followed by Dunnett's post hoc test. *$P < 0.05$, ***$P < 0.001$. Scale bars, 100 μm (**c**, **g**) 300 μm (**d**, **h**).

metronidazole, showed effects similar to those of doranidazole on mitochondrial complex protein expression (Supplementary Fig. 4e). Consistent with the marked decrease in basal OCR, a 12-h doranidazole treatment significantly reduced the ATP content of GSC-H cells under mild hypoxic conditions (Fig. 4f).

**Doranidazole changes intracellular metabolite levels in GSCs.** Both ferroptosis[28] and mitochondrial function are intimately linked to cellular metabolism. We therefore examined the effects of doranidazole on GSC metabolism by capillary electrophoresis coupled with mass spectrometry. Intermediates of the

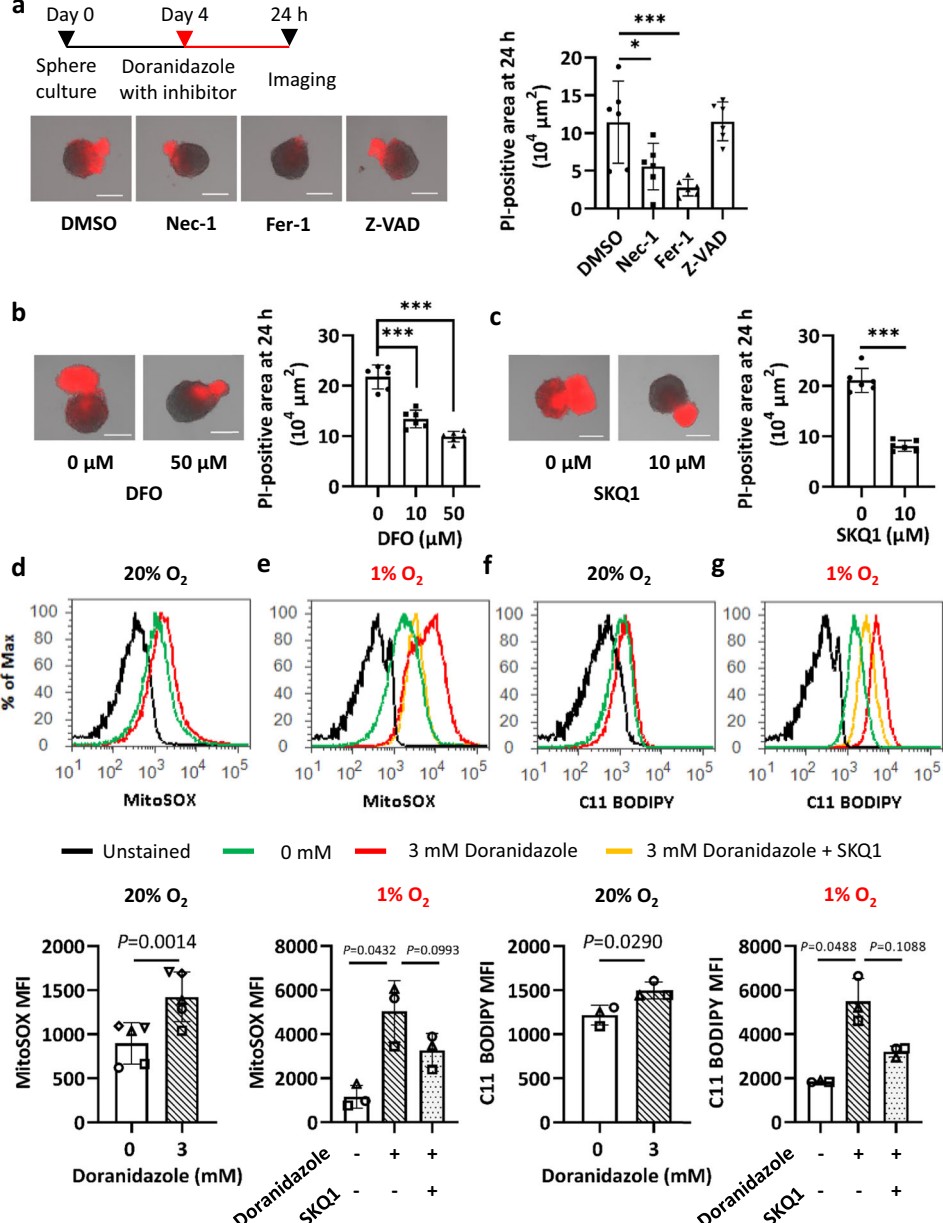

**Fig. 3 Mechanism of doranidazole-induced GSC death. a** Evaluation of cell death based on PI uptake for GSC-H spheres incubated for 24 h with 3 mM doranidazole and either DMSO vehicle or the cell death inhibitors (100 μM) necrostatin-1 (Nec-1), ferrostatin-1 (Fer-1), or Z-VAD-FMK (Z-VAD). Representative images and quantification of the PI-positive sphere area are shown. **b** Evaluation of cell death as in (**a**) for GSC-H spheres incubated for 24 h with 3 mM doranidazole and the indicated concentrations of deferoxamine (DFO). **c** Evaluation of cell death as in (**a**) for GSC-H spheres incubated for 24 h with 3 mM doranidazole and SKQ1 (0 or 10 μM). Flow cytometric analysis of MitoSOX Red staining in GSC-H cells incubated with or without 3 mM doranidazole and 10 μM SKQ1 under normoxic (**d**) or hypoxic (**e**) conditions for 12 h. Representative profiles as well as quantification of the mean fluorescence intensity (MFI) of MitoSOX Red from $n = 5$ (**d**) or $n = 3$ (**e**) independent experiments are shown. **f, g** Flow cytometric analysis of BODIPY 581/591 C11 staining in GSC-H cells incubated with or without 3 mM doranidazole and 10 μM SKQ1 under normoxic (**f**) or hypoxic (**g**) conditions for 18 h. Representative profiles and quantification of the MFI of BODIPY 581/591 C11 from $n = 3$ independent experiments are shown. Quantitative data are means ± s.d. from six biologically distinct replicates in a representative experiment, $n = 3$ independent experiments performed **a–c** or from the indicated number of independent experiments **d–g** and were analyzed by one-way ANOVA followed by Dunnett's post hoc test **a**, **b**, the unpaired two-tailed Student's $t$-test (**c**), the paired two-tailed Student's $t$-test **d**, **f**, or one-way ANOVA followed by Tukey's post hoc test **e**, **g**. *$P < 0.05$, ***$P < 0.001$. Scale bars, 300 μm (**a–c**).

tricarboxylic acid (TCA) cycle showed the most marked changes in abundance after exposure of GSC-H cells to doranidazole. The intracellular levels of citrate, *cis*-aconitate, and succinate were increased, whereas those of fumarate and malate were decreased, by exposure of the cells to doranidazole for 24 h (Fig. 5a), consistent with changes resulting from a decrease in the activity of mitochondrial complex II. The amounts of glutamine, glutamate,

and α-ketoglutarate were also significantly reduced by doranidazole treatment, suggestive of perturbation of glutaminolysis. Of note, the ratio of NADH to $NAD^+$ was significantly increased by doranidazole treatment (Fig. 5b). Considering that elevated levels of $NADH/NAD^+$ inhibit dehydrogenases in the TCA cycle, and that increased succinate suggests a blockade of complex II, these alterations in TCA metabolites appear to result from

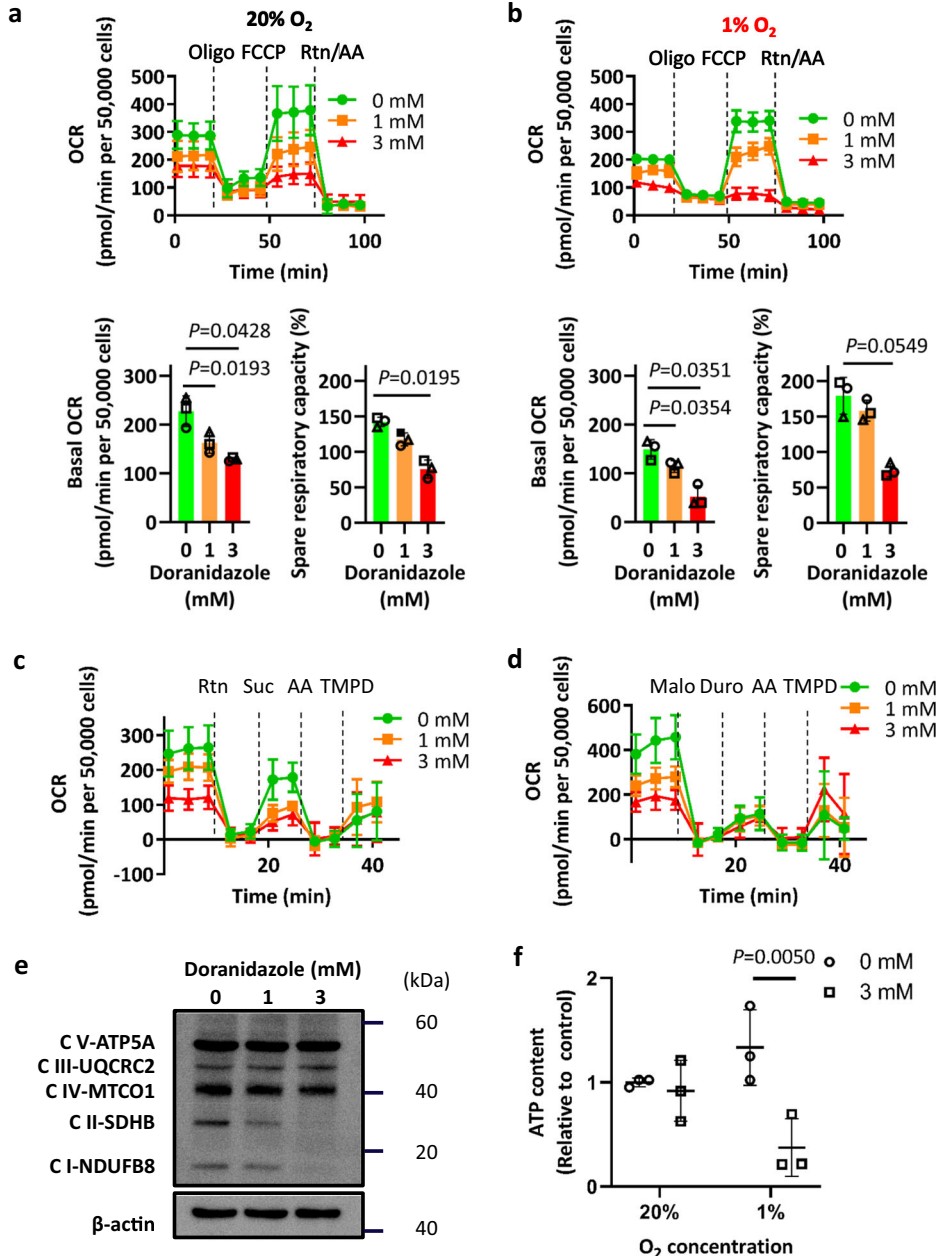

**Fig. 4 Effect of doranidazole on mitochondrial function.** Extracellular flux analysis of GSC-H cells treated for 12 h with the indicated concentrations of doranidazole under normoxic (**a**) or hypoxic (**b**) conditions. Representative examples of changes in OCR after sequential injection of the indicated inhibitors (Oligo oligomycin, Rtn rotenone, AA antimycin A) as well as quantitative data for basal OCR and spare respiratory capacity (means ± s.d. for $n = 3$ independent experiments) are shown. Mitochondrial complex I-, complex II-, and complex IV-dependent OCR (**c**) and complex II-, complex III-, and complex IV-dependent OCR (**d**) measured in GSC-H cells after exposure to the indicated concentrations of doranidazole for 12 h and permeabilization of the plasma membrane. Suc succinate, Malo malonate, Duro duroquinol. Data are means ± s.d. for at least five biologically distinct replicates in $n = 1$ experiment. **e** Immunoblot analysis of mitochondrial complex (C) proteins in GSC-H cells treated for 24 h with the indicated concentrations of doranidazole. **f** ATP content of GSC-H cells exposed to doranidazole under normoxic or hypoxic conditions for 12 h. Data are expressed relative to the value for normoxia and 0 mM doranidazole and are means ± s.d. from $n = 3$ three independent experiments. Statistical analysis was performed by one-way ANOVA followed by Dunnett's post hoc test **a**, **b**, or with the two-way ANOVA followed by Sidak's post hoc test **f**.

impairment of mitochondrial respiration. Furthermore, while a 12-h treatment with doranidazole was insufficient to induce robust changes in ATP in normoxic conditions (Fig. 4f), a 24-h treatment significantly decreased ATP and increased AMP contents (Fig. 5c). These effects coincided with a significant but modest drop of the adenylate charge (Fig. 5d). In contrast to doranidazole, metronidazole had a negligible effect on metabolites overall (Supplementary Fig. 5a–d).

**Doranidazole limits GSC growth in cultured brain slices.** Finally, to assess the effects of doranidazole on GSCs in their niche, we first examined cultured brain slices from mice with orthotopic implants of GSC-F cells. Such brain slices allow the visualization of glioma cells in their native environment, without the limitations imposed by a functional blood–brain barrier[35]. Under normoxic conditions, GSC-F cells in brain slices incubated with 3 mM doranidazole for 2 days gradually accumulated in $G_1$

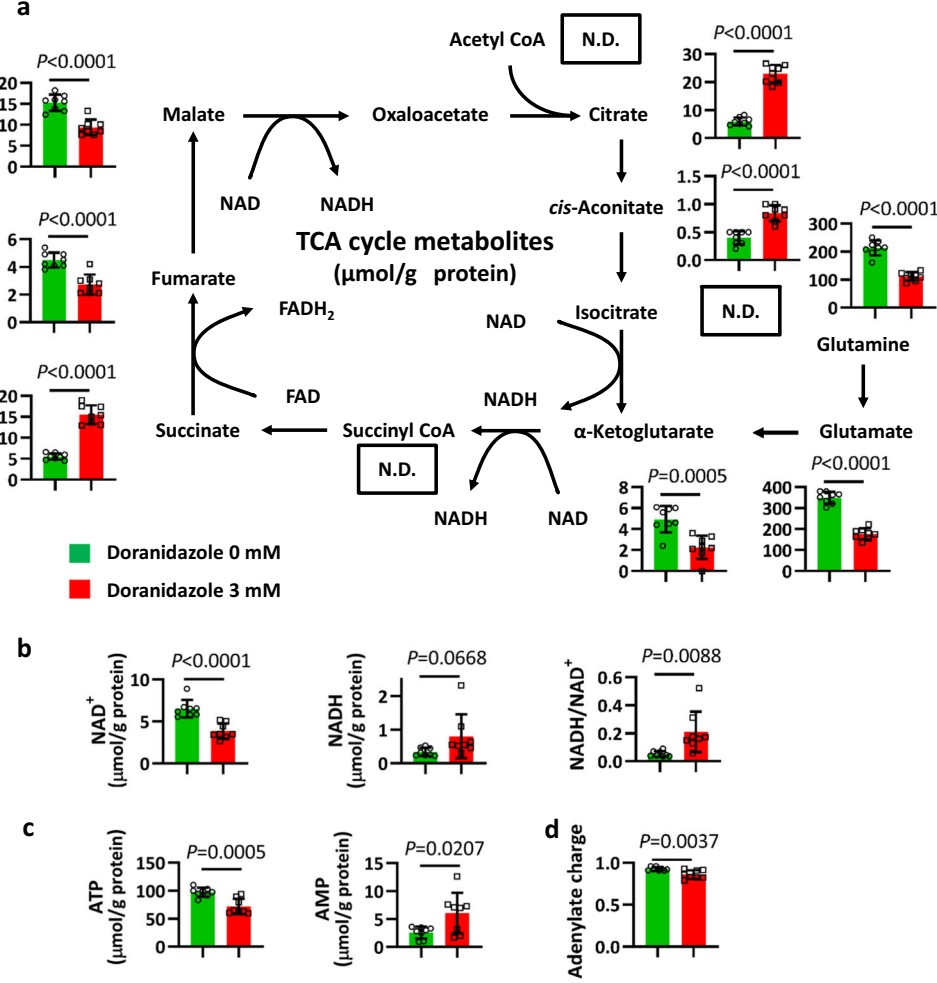

**Fig. 5 Effects of doranidazole on intracellular metabolite levels in GSC-H cells. a** Metabolome analysis for the TCA cycle in GSC-H cells exposed to 0 or 3 mM doranidazole for 24 h. N.D. not detected. The amounts of $NAD^+$ and NADH and the $NADH/NAD^+$ ratio (**b**) as well as ATP and AMP (**c**) and the total adenylate charge (**d**) were also determined. Metabolite levels were normalized by total protein amount, are means ± s.d. from eight biologically distinct replicates, $n = 1$ experiment, and were analyzed by the unpaired two-tailed Student's $t$-test.

phase of the cell cycle (Fig. 6a) and tumor growth was inhibited (Fig. 6c). Given that brain slices containing large hypoxic tumors are less amenable to imaging, we next exposed slices with similar small-sized tumors to 1% $O_2$. Under this condition, doranidazole induced cell death, as revealed by the binding of 4′,6-diamidino-2-phenylindole (DAPI) to DNA and by morphological changes apparent on hematoxylin-eosin staining (Fig. 6b, c). The effects of doranidazole under both normoxic and hypoxic conditions were concentration dependent (Fig. 6a, b and Supplementary Fig. 6a, b).

Next, to reproduce and examine early and pronounced intratumoral hypoxic regions, we injected mice subcutaneously with GSC-H cells. Doranidazole (200 mg/kg, i.p.) administration for 5 days delayed the growth of tumors formed by cells injected 15 days previously (Supplementary Fig. 6c) as well as reduced the EF5-positive hypoxic tumor area (Supplementary Fig. 6d), although this latter effect was not statistically significant. Together, these results thus showed that doranidazole induced GSC death in hypoxic intratumoral regions in both the ex vivo and in vivo settings.

## Discussion

We have here shown that 2-nitroimidazole drugs sensitize hypoxic murine GSCs to ionizing radiation in association with an

increase in the number of double-strand breaks. Furthermore, under hypoxic conditions, these drugs manifest radiation-independent cytotoxicity for GSCs associated with the induction of mitochondrial dysfunction and ferroptotic cell death. The decrease in the number of metabolically active GSCs induced by these agents is likely to reduce the intratumoral competition for oxygen and thereby ultimately to limit the size of the hypoxic niche.

Nitroimidazoles have previously been shown to potentiate radiation-induced DNA damage and the subsequent death of actively cycling cancer cells in several tumor types[36]. However, cancer stem cells often possess an enhanced ability to repair DNA damage and cycle more slowly compared with differentiated tumor cells[14,37,38]. We have now shown that doranidazole potentiated the radiation-induced increase in the number of double-strand breaks in mouse GSCs and served as a radio-sensitizer for GSC-derived tumors. At least for GBM, these findings exclude the possibility that intrinsic resistance of cancer stem-like cells to 2-nitroimidazoles is a major cause of the limited radiosensitization observed in clinical studies[39–41] and suggest the need for an increased focus on theranostic applications.

GBMs are thought to contain substantial hypoxic regions, to experience loss of blood–brain barrier function, and to be highly recalcitrant to the effects of chemo- and radiotherapy. In particular, large tumors presenting with contrast enhancement and

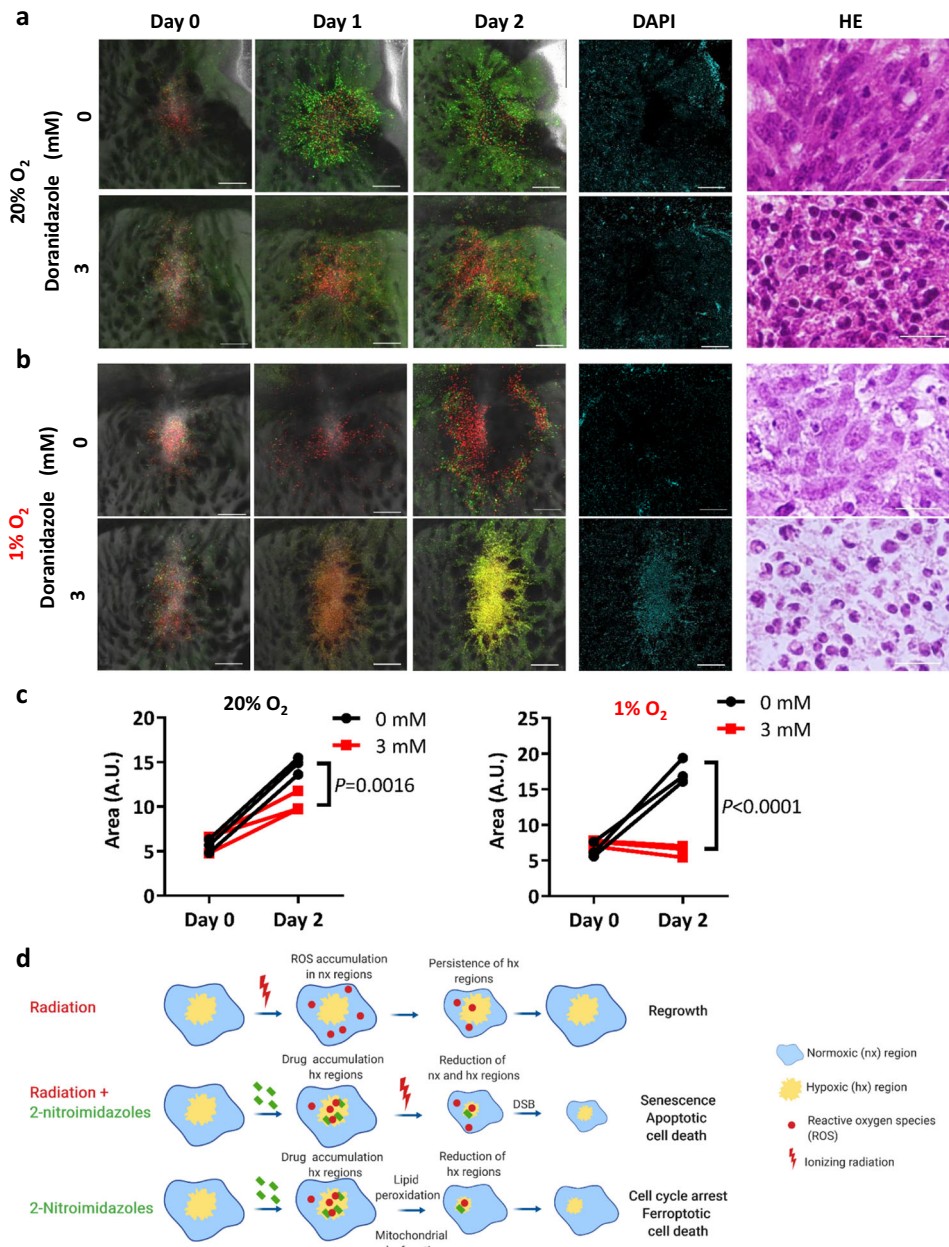

**Fig. 6 Effects of doranidazole on cultured brain slices bearing implants of GSC-F cells.** Sequential images of brain slices prepared from mice bearing GSC-F cell–based orthotopic tumors and incubated with or without 3 mM doranidazole under normoxic (**a**) or hypoxic (**b**) conditions. Overlays of green fluorescence (S–G$_2$–M phase), red fluorescence (G$_1$ phase), and phase-contrast images are shown for days 0, 1, and 2. A DAPI exclusion assay for the same areas as well as hematoxylin-eosin (HE) staining of paraffin-embedded sections of the explants are also presented for day 2. Scale bars, 300 μm (fluorescence images) or 50 μm (hematoxylin-eosin). **c** Quantification of tumor area for explants as in (**a, b**) at days 0 and 2. Data are for explants from $n = 3$ mice per condition and were analyzed by two-way ANOVA followed by Sidak's post hoc test. A.U. arbitrary units. **d** Model for the effects of doranidazole on GSCs in their hypoxic niche. DSB double-strand breaks. The scheme was created with biorender.com.

compression symptoms as well as recurrent tumors usually contain not only hypoxic regions but also an increased number of GSCs, and such tumors might therefore benefit from doranidazole treatment. The identification of such tumors would be an essential step in any future clinical applications of this drug. A high initial value followed by a gradual decrease in [$^{18}$F]fluoromisonidazole uptake by GBM tumors could indicate treatment efficacy, with such imaging thus representing a companion diagnostic option. In the case of other solid tumors, stratification of patients based on [$^{18}$F]fluoromisonidazole uptake might be necessary for maximal exploitation of the radiosensitizing potential of 2-nitroimidazoles.

Monitoring of hypoxia would also be essential for clinical exploitation of the direct cytotoxicity of doranidazole for GSCs. We found that doranidazole, misonidazole, and etanidazole each induced pronounced radiation- and cell cycle-independent cell death within the oxygen-deprived center of GSC spheres. In spheres with a diameter of >500 μm, which already have a small core of dead cells as a result of oxygen and nutrient limitation, GSC death was apparent as early as 12 h after initiation of drug treatment. Such cell death was markedly, but not completely, attenuated by the ferroptosis inhibitor ferrostatin-1, the iron-chelator deferoxamine, and the mitochondrial ROS scavenger SKQ1, whereas it was not affected by the caspase inhibitor Z-

VAD-FMK. These results suggest that the GSC death induced by 2-nitroimidazoles is mediated largely by ferroptosis rather than by apoptosis.

Ferroptosis is a form of iron-dependent oxidative cell death mediated by ROS accumulation and lipid peroxidation[28,42]. The relation between mitochondrial ROS generation and ferroptosis remains poorly understood but seems to be context dependent[28]. Partial attenuation of ferroptosis by oligomycin[43], an inhibitor of mitochondrial complex V, implicates the electron transport chain as a possible initial source of ROS. In contrast, the detection of ferroptosis in cells that lack mitochondrial DNA or mitochondrial function has led to the suggestion that changes in mitochondrial morphology and membrane potential associated with ferroptosis represent the final stages of the death cascade[44]. We have now found that doranidazole induced a decrease in mitochondrial complex I and II activity, an increase in the NADH/NAD$^+$ ratio, and the accumulation of succinate in GSCs. Together with the partial rescue of doranidazole-induced GSC death by SKQ1, these results suggest that the accumulation of mitochondrial ROS, even though due to aberrant function of complexes I and II[34], takes place during the reversible phase of ferroptosis and thus closer to the initiating rather than the end stage (Fig. 6d). Of note, the decrease in mitochondrial complex activity, the increase in ROS and lipid peroxidation, and the changes in the transcriptome induced by doranidazole were qualitatively similar in normoxic and mildly hypoxic conditions but tended to be more pronounced in the latter. The degree of ferroptosis will likely be maximal under conditions in which oxygen tension is sufficiently low to allow the intracellular accumulation of nitroimidazoles, but sufficiently high to allow cells to rely on mitochondrial and electron transport chain function and to favor a net increase in ROS levels.

The net oxidative damage will also depend on the antioxidant responses triggered by the treatment in a cell-specific manner. In GSCs, doranidazole increased the expression of several antioxidant-related genes, including the aldehyde dehydrogenase gene *Aldh3a1*. Aldh3a1 protects cells by detoxifying lipid peroxidation-derived reactive aldehydes such as 4-hydroxynonenal[45]. Aldh3a1 can also release antioxidant nitric oxide (NO) from S-nitrosothiols[45–47]. Upregulation of Aldh3a1 has been identified as a cause of resistance to ferroptosis induced by sulfasalazine[48], an inhibitor of the cystine–glutamate transporter subunit xCT, and drug screening for synthetic lethality identified aldehyde dehydrogenase inhibitors that cooperatively enhance the effects of sulfasalazine on cell lines derived from several types of solid tumors as well as on subcutaneous tumors in animal models[48,49]. The upregulation of antioxidant proteins might thus be another reason why nitroimidazoles have failed to realize their expected clinical potential. Screening for synthetic lethality under hypoxic conditions might identify aldehyde dehydrogenase inhibitors that specifically enhance the action of nitroimidazoles.

Differences in electron affinity and reduction potential between nitroimidazoles appear to play a crucial role in explaining the effects of doranidazole in ferroptosis, since those characteristics have been found to be correlated with hypoxia selectivity and toxicity[50]. Consistent with these previous findings, we found that equimolar concentrations of 2-nitroimidazoles doranidazole, misonidazole, and etanidazole (electron affinity values of more than −400 mV[15,50]), but not of 5-nitroimidazoles metronidazole and nimorazole (electron affinity values of <−400 mV), were able to induce hypoxic GSC death. In our previous study using the same GBM model, we reported upregulation of nucleophilic polysulfides in tumors[51], and further investigation is necessary to determine whether the electrophilic doranidazole breaks the balance between endogenous aldehydes and polysulfides to render GSCs more susceptible to cell death.

In parallel with mitochondrial dysfunction, mobilization of the intracellular labile iron pool is a major determinant of ferroptosis. Doranidazole treatment upregulated *Steap3* and *Hmox1* in GSCs in both normoxic and mildly hypoxic conditions. Metallor-eductase Steap3 catalyzes the reduction of ferric iron (Fe$^{3+}$) to ferrous one (Fe$^{2+}$)[30]. Heme oxygenase 1 increases the labile iron pool by releasing Fe$^{2+}$ from heme[52] and is instrumental in era-stin- and withaferin A-induced ferroptosis[31,52]. Furthermore, NADH, which was increased after doranidazole treatment, can also contribute to the reduction and mobilization of Fe$^{2+}$ from ferritin, especially under anaerobic conditions[53].

An issue that will require further consideration is the involvement of ferroptosis in 2-nitroimidazole-induced normal tissue toxicity. We found that doranidazole induces a similar extent of cell death in NSCs differentiated along astrocytic lines as it does in the GSCs in conditions of <0.1% O$_2$. Given that in the human brain, normal cells are expected to reside at physiological oxygen concentrations of 0.5–7%[25], massive cell death due to accumulation of the drugs in severely hypoxic normal cells is less likely to occur. However, although not specifically addressed in this study, the doranidazole-induced mitochondrial dysfunction and sub-lethal damage to oxic GSCs could also occur in non-transformed cells and might partially contribute to the toxicities reported for 2-nitroimidazoles in normal tissues.

In conclusion, we have demonstrated the potential of 2-nitroimidazoles as targeted inducers of ferroptosis in hypoxic GSCs within their niche. Advances in knowledge of cancer cell hierarchies and types of cell death have thus prompted further exploration of 2-nitroimidazole compounds that has led to the uncovering of previously unknown aspects of their reductive metabolism and suggested possible future theranostic applications.

## Methods

**Establishment of induced GICs and GSCs and cell culture**. Primary *Ink4a/Arf*-null NSCs were isolated from the subventricular zone of 6-week-old mice as previously described[23]. NSCs and all derived cells were cultured in neural stem cell medium (NSM), consisting of Dulbecco's modified Eagle's medium (DMEM)-F12 (Wako, Osaka, Japan) without serum and containing recombinant human epidermal growth factor and basic fibroblast growth factor (PeproTech, Rocky Hill, NJ) at 20 ng/ml, heparan sulfate (Sigma-Aldrich, St. Louis, MO) at 200 ng/ml, and B27 supplement without vitamin A (Invitrogen, Carlsbad, CA). For establishment of GICs, retrovirus-containing culture supernatants were prepared as previously described[23] from the pBABE-Hygro retroviral vector containing human *H-Ras$^{V12}$* cDNA (kindly provided by P. P. Pandolfi and T. Maeda). NSCs were infected with the retrovirus-containing supernatants, and the infected cells were purified by selection with hygromycin (200 μg/ml) for 14 days. The resulting cells were designated GIC-H[54]. Primary tumors formed by orthotopic implantation of GIC-H cells in syngeneic mice were excised at the onset of tumor-related symptoms. Dissociated tumor cells were subjected to selection with hygromycin (200 μg/ml) for 14 days, and the resulting cells were designated GSC-H and maintained in NSM. The human glioma cell lines U251MG (GBM) and Becker (astrocytoma grade III–IV) were obtained from American Type Culture Collection and Japanese Collection of Research Bioresources Cell Bank, respectively, and they were maintained in DMEM-F12 supplemented with 10% FBS before a switch to NSM for the assays in this study. For induction of cell differentiation, NSCs or GICs were cultured in DMEM-F12 supplemented with 1% or 10% FBS for 72 h. All cells were studied within 20 passages after establishment.

**Chemicals and reagents**. Doranidazole was obtained from POLA Pharma Inc. (Tokyo, Japan) and dissolved in phosphate-buffered saline (PBS). Metronidazole and misonidazole (Sigma-Aldrich) as well as etanidazole (Santa Cruz Biotechnology, Dallas, TX) and nimorazole (Aobious, Gloucester, MA) were dissolved in dimethyl sulfoxide (DMSO). Ferrostatin-1, DAPI, and NAC were obtained from Sigma-Aldrich, necrostatin-1 from Abcam (Cambridge, UK), Z-VAD-FMK from MBL (Nagoya, Japan), deferoxamine mesylate from Tokyo Chemical Industry (Tokyo, Japan), SKQ1 from Cayman Chemical (Ann Arbor, MI), MG132 and EF5 from Merck Millipore (Billerica, MA), and Hoechst 33342 from Thermo Fisher Scientific Inc. (Waltham, MA).

**Irradiation**. X-irradiation was performed with the use of an MBR-1520R-4 system (Hitachi Power Solutions, Ibaraki, Japan) at settings of 150 kV and 20 mA. The dose rate of radiation was 1.45 Gy/min.

**γH2AX staining and imaging**. Cells were incubated with drug under normoxic or hypoxic conditions for 30 min, exposed to 0 or 2 Gy of ionizing radiation, and fixed 30 min later with 4% paraformaldehyde. They were then stained consecutively with rabbit polyclonal antibodies to γH2AX (phospho-S139, Abcam, 1:1000 dilution) and Alexa Fluor 488-conjugated secondary antibodies (Thermo Fisher Scientific Inc.). Nuclei were counterstained with Hoechst 33342. Images were acquired with a Fluoview FV10i confocal microscope and were uniformly processed with Fluoview software v4.2 (Olympus, Tokyo, Japan). The number of immunoreactive foci was counted with the use of ImageJ software and the plugin PzFociEZ[55]. In brief, a nuclear region of interest was generated from Hoechst 33342 staining, and the number of foci in this region was automatically counted for γH2AX staining. Background subtraction was performed with the rolling ball algorithm. The radius of the rolling ball was set at 50 pixels and Subtract Value set to 5. The number of foci was counted with the use of FindMaxima at a noise tolerance of 7. At least 50 cells were analyzed in each experiment, and three independent experiments were performed.

**Hypoxia induction**. For irradiation under hypoxic culture conditions, a BIONIX-3 hypoxic culture kit (Sugiyama-Gen, Tokyo, Japan) was used to obtain an $O_2$ concentration of <0.1%. Cells were irradiated 3 h after pouch sealing, and the $O_2$ concentration was monitored during the experiments. For experiments not including irradiation, cells were cultured under 1% $O_2$ with nitrogen replacement in a multigas incubator or under <0.1% $O_2$ after pouch sealing at 37 °C. Experiments were performed under normoxic conditions when not indicated otherwise.

**Colony formation assay**. For evaluation of the surviving fraction after irradiation, cells were seeded in 10-cm dishes. On the basis of preliminary experiments to determine the cell numbers necessary to compensate for plating efficiency, cells were plated at densities of $1 \times 10^3$, $2 \times 10^3$, $5 \times 10^3$, or $1 \times 10^5$ for the 0, 2, 5, and 10 Gy treatment groups, respectively. They were allowed to settle for 2 h before treatment. Doranidazole was added 3 h before irradiation, and the cells were incubated under normoxic or hypoxic conditions. After irradiation, the medium was replaced with fresh NSM and the cells were cultured under normoxic conditions for the remainder of the experiment. Ten days after plating, the cells were fixed with 4% paraformaldehyde and stained with toluidine blue O (Sigma-Aldrich). Images of individual plates were obtained, and colonies were counted with the use of the Analyze Particles Function of ImageJ software. Three independent experiments were performed.

**Sphere growth assay**. GIC-H or GSC-H cells were manually plated in low-binding 96-well plates (Corning, Corning, NY) at a density of 100 cells per well and were treated with 0, 2, 4, 6, or 8 Gy. Images of the cells were acquired with a Biorevo BZ-9000 inverted microscope (Keyence, Osaka, Japan) at 10 days after plating. Sphere area was quantified with Keyence Analysis Software.

**Flow cytometric analysis of cell death**. Cell death was analyzed by flow cytometry with the use of PI[56]. Cells incubated with drug for 1 or 3 days were collected and exposed to PI (1 µg/ml) before flow cytometric analysis with an Attune flow cytometer (Thermo Fisher Scientific).

**Visualization of cell death and hypoxic regions in spheres**. For evaluation of cell death, cells were manually plated in low-binding 96-well plates (Corning) at a density of 100 cells per well. Drugs, inhibitors, and PI (0.2 µg/ml) were added after spheres had achieved diameters of ~500 µm, and the cells were then incubated for a further 24 h. Images were acquired with a Biorevo BZ-9000 inverted microscope (Keyence), and the PI-positive-area (area containing dead cells) of each sphere was quantified with Keyence Analysis Software. For evaluation of sphere hypoxia, GSC-H cells were manually plated at a density of 100 cells per well and cultured for 5 days. The resulting spheres were incubated in the presence of 200 µM EF5[57] for 4 h, fixed with 4% paraformaldehyde, embedded in paraffin, and sectioned at a thickness of 4 µm. The sections were stained with the use of an EF5 Hypoxia Detection Kit (Merck Millipore), and nuclei were counterstained with Hoechst 33342. U251MG and Becker cells were plated at a density of 10,000 cells per well and cultured for 2 days before evaluation of sphere hypoxia in the same manner.

**Animal experiments**. All animal experiments were approved by the Animal Care and Use Committee of Keio University School of Medicine. Orthotopic implantation of cells was performed as described previously[23]. In brief, female C57BL/6 J mice (age 6–8 weeks) were anesthetized and placed into a stereotactic apparatus (David Kopf Instruments, Tujunga, CA). One thousand viable GSC-H cells were injected into the right hemisphere at a position 2 mm lateral to the bregma and 3 mm below the brain surface. After 10 days, animals were exposed to 15 Gy (radiation dose rate, 1.45 Gy/min) with or without prior injection of doranidazole (200 mg/kg, i.p.). Radiation was confined to the brain by protection of the body with a lead shield. Animals were monitored for the development of neurological deficits and weight loss. Brain slice explants were established as described previously[35]. In brief, $1 \times 10^4$ viable GSC-F cells were injected into the right hemisphere. After 7 days, the brain was removed and cut into 200-µm-thick coronal

slices with the use of a VS1200 vibratome (Leica, Wetzlar, Germany). The explants were cultured on Millicell-CM culture inserts (Merck Millipore) in glass-bottom dishes and were maintained in the presence of drug in NSM under normoxic or hypoxic conditions. Images were acquired with an FV10i confocal microscope (Olympus) and uniformly processed with the Olympus Fluoview Software v4.2b. Tumor area was contoured manually and calculated with the use of ImageJ. For evaluation of the effect of doranidazole on subcutaneous tumors, $5 \times 10^5$ viable GSC-H cells were injected subcutaneously into the flank of female C57BL/6J mice. After 15 days, doranidazole was injected i.p. at a dose of 200 mg/kg for 5 days. Control animals received vehicle only. At 20 days after cell implantation, 10 mM EF5 (volume in milliliters equal to 1/100 of body weight in grams) was injected i.p., and tumors were excised 4 h later and measured with calipers. Tumor volume was calculated[58] as (width$^2 \times$ length)/2. The tumors were then fixed with 4% paraformaldehyde, embedded in paraffin, and sectioned at a thickness of 4 µm. The sections were depleted of paraffin before staining with Alexa Fluor 488-conjugated antibodies to EF5 (EF5 Hypoxia Detection Kit, Merck Millipore) at a concentration of 75 µg/ml. Nuclei were counterstained with Hoechst 33342. Images were acquired with an FV10i confocal microscope (Olympus) and processed in a uniform manner with the Olympus Fluoview Software v4.2b. EF5 staining is presented in orange. The EF5-positive area was determined as a percentage of the total tumor (Hoechst 33342-positive) area with the use of ImageJ.

**Measurement of ATP production**. Cells incubated with drug for 12 h were isolated to obtain a single-cell suspension in PBS, washed, and plated at a density of $5 \times 10^4$ per well in 96-well plates in the absence of drug for 30 min. Cellular ATP content was determined with the use of a CellTiter-Glo kit (Promega, Madison, WI) and microplate reader (Perkin Elmer, Waltham, MA).

**Measurement of OCR**. OCR was determined with the use of a Seahorse XF Extracellular Flux Analyzer (Agilent, Santa Clara, CA). In brief, dissociated cells were suspended in NSM containing drug, plated in 24-well plates (Agilent) that had been coated with Matrigel diluted 1:10 in Seahorse XF Assay Medium (Agilent), and incubated under normoxic or hypoxic conditions for 12 h. The cells were then incubated in Seahorse XF Assay Medium supplemented with 17.5 mM glucose, 2 mM pyruvate, and 2.5 mM glutamine for 1 h. For evaluation of mitochondrial function, cells were exposed sequentially to metabolic inhibitors: 1 µM oligomycin (inhibitor of ATP synthase), followed by 4 µM FCCP (uncoupler of mitochondrial oxidative phosphorylation), followed by a combination of 500 nM rotenone (inhibitor of mitochondrial complex I) and 500 nM antimycin A (inhibitor of mitochondrial complex III). Basal OCR and changes induced by the metabolic inhibitors were measured, and values were normalized by cell number at the end of the experiment.

For analysis of mitochondrial complex activity, cells were incubated in NSM with or without doranidazole, after which the medium was replaced with Mitochondria Assay Solution[59]. The plasma membrane was permeabilized to allow nonpermeable substrates to access mitochondria with the use of XF Plasma Membrane Permeabilizer (Agilent). The activity of individual complexes was assayed with combinations of specific substrates and inhibitors[59]: for assay of complex I-dependent respiration, 10 mM pyruvate and 2.5 mM malate as substrates and 2 µM rotenone as an inhibitor; for assay of complex II-dependent respiration, 10 mM succinate as substrate and 2 µM rotenone (complex I activity) followed by 10 mM malonate (complex II activity) as inhibitors; for assay of complex III-dependent respiration, 0.5 mM duroquinol as substrate and 2 µM antimycin A as inhibitor; and for assay of complex IV-dependent respiration, 0.5 mM TMPD and 2 mM ascorbate as substrates.

**Immunoblot analysis**. Cells were passed repeatedly through a 27-gauge needle in radioimmunoprecipitation buffer (Sigma-Aldrich) containing PhosSTOP and cOmplete Mini phosphatase and protease inhibitor cocktails (Sigma). The resulting extracts were fractionated by SDS-polyacrylamide gel electrophoresis, and the separated proteins were transferred to a polyvinylidene difluoride membrane (Bio-Rad, Hercules, CA) with the use of a Trans-Blot Turbo Transfer Starter System (Bio-Rad). Immunoblot analysis was performed with the primary antibodies listed in Supplementary Information.

**Measurement of mitochondrial superoxide production**. Mitochondrial superoxide production was measured by flow cytometry with the use of MitoSOX Red Mitochondrial Superoxide Indicator (Thermo Fisher Scientific Inc.). Cells were incubated with drug under normoxic or hypoxic conditions for 12 h. After the addition of MitoSOX Red to a final concentration of 5 µM, the cells were incubated for an additional 10 min and then analyzed with an Attune flow cytometer (Thermo Fisher Scientific Inc.). Mean fluorescence intensity (MFI) was calculated with Attune software v2.1.0 (Thermo Fisher Scientific Inc.).

**Measurement of lipid peroxidation**. Lipid peroxidation was measured by flow cytometry with the use of BODIPY 581/591 C11 (Thermo Fisher Scientific). Cells were incubated with drug under normoxic or hypoxic conditions for 18 h. After the addition of BODIPY 581/591 C11 to a final concentration of 5 µM, the cells were incubated for an additional 30 min, isolated by exposure to trypsin, stained for

5 min with PI in DMEM for dead cell exclusion, and then analyzed with an Attune flow cytometer (Thermo Fisher Scientific Inc.) with excitation at 488 nm and a 530/30-nm emission filter to detect oxidized forms of the probe. MFI was calculated with Attune software v2.1.0 (Thermo Fisher Scientific Inc.).

**Metabolome analysis**. Cells were plated at a density of $2 \times 10^5$ per 10-cm dish and cultured for 2 days, after which the medium was replaced and the cells were incubated for 24 h with drug, washed with 10% mannitol (Wako), and exposed to ice-cold methanol containing internal standards of methionine sulfone and 2-morpholinoethanesulfonic acid. Intracellular metabolites were quantified by capillary electrophoresis–mass spectrometry with an Agilent CE system as described previously[51,60]. A total of eight biologically distinct replicates was prepared for each group, and metabolite levels were normalized by total protein amount.

**Statistics and reproducibility**. Statistical analysis was performed with GraphPad Prism software. A $P$ value of <0.05 was considered statistically significant. For animal experiments, survival curves were compared with the log-rank test. For experiments using GSCs in culture, quantitative data compare means ± s.d. obtained from the indicated number ($n$) of independent experiments. Within each independent experiment, biologically distinct replicates were used to generate averages for each condition as described in the figure legend. Data were compared with the paired or unpaired two-tailed Student's $t$-test, by one-way ANOVA followed by Dunnett's or Tukey's post hoc tests or by two-way ANOVA followed by Sidak's post hoc test, as appropriate. Markers are used to distinguish paired data in the graphs.

**Reporting summary**. Further information on research design is available in the Nature Research Reporting Summary linked to this article.

## Data availability

All source data underlying the graphs presented in the main and supplementary figures are provided in the Supplementary Information file "Source data". Uncropped images of the immunoblots and gating information for the flow cytometry analyses are presented in the "Supplementary Information" file. The microarray data was deposited in the Gene Expression Omnibus (GEO) database and can be accessed under GSE135858. Other data supporting the findings of this study are available from the corresponding author upon reasonable request.

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

## Acknowledgements

We thank I. Ishimatsu for preparing the histopathology samples, M. Sato and M. Kobori for help in preparation of the paper, and Collaborative Research Resources of Keio University School of Medicine for technical assistance. Funding: This work was supported by KAKENHI Grants-in Aid for Scientific Research (C) to O.S. (nos. 16K07124 and 19K07671) and a Grant-in Aid for Early Career Scientists to N.K. (no. 18K15602) from the Ministry of Education, Culture, Sports, Science, and Technology of Japan.

## Author contributions

N.K., O.S., and H.S. designed the study. N.K. and O.S. analyzed the data and wrote the paper. N.K. acquired most of the experimental data. Y.N., N.H., T.M., and T.H. performed the metabolome analysis. N.K., R.K. and J.F. performed irradiation experiments. N.K. and N.O. constructed the bicistronic vector. M.S. was the leader of JST ERATO Suematsu Gas Biology until March 2015 who established the infrastructure for metabolomics analysis used in this study. N.S., M.S., and H.S. supervised the project.

## Competing interests

H.S. has received commercial research grants from POLA Pharma Inc. and Nihon Noyaku Co. Ltd. O.S. has received research support from Nihon Noyaku Co. Ltd. The findings of this study are the subject of a Japanese patent application as follows: Applicants: Keio University and POLA Pharma Inc. Inventors: H.S., O.S., N.K., and N. Kubota. Application no.: PCT/JP2018/036792. Status: international publication. Specific aspects of the study covered in the application: radiosensitizing effect of doranidazole on GSCs as well as its effects on mitochondrial complexes and ROS. Doranidazole used in this study was provided by POLA Pharma Inc. All other authors declare no competing interests.
