## [Peer Review File · Communications Biology]

Reviewers' comments:

Reviewer #1 (Remarks to the Author):

This is a data-rich, comprehensive and well-presented study that explores the mechanism of cytotoxicity of the 2-nitroimidazoles doranidazole and misonidazole. The evidence that ferroptosis contributes to cell killing is novel, but I have a number of significant concerns about the manuscript:

1. The first sentence of the Abstract is misleading. Radiosensitisation by nitroimidazoles is due to the parent drug, not their reduced metabolites. This error continues in the second sentence, and in the first paragraph of the Introduction. Second paragraph of the Introduction shows that the authors do understand the mechanisms behind the three reported properties of nitroimidazoles (radiosensitisation, hypoxia-selective cytotoxicity, hypoxia imaging), but the Abstract and first para need to be redrafted to make this clear.

2. The above notwithstanding, there is no clear evidence that any 2-nitroimidazole (including doranidazole) has therapeutic potential as a hypoxic cell cytotoxin (as distinct from activity as radiosensitisers). In that respect the mechanism of cytotoxicity, which is the focus of this study, is of limited interest, although it could be relevant to the well-studied neurotoxicity of 2-nitroimidazoles (which is not mentioned by the authors).

3. The authors report a surprising lack of radiosensitisation by doranidazole in vitro (Fig. 1F,I), even under severe hypoxia, despite the use of a very high concentration of the drug (6 mM). They claim statistically significant radiosensitisation, but I fail to see how they have determined such low p values. Anyway, the effect is biologically insignificant even if it is statistically significant. This is difficult to understand, as is the absence of hypoxic selectivity of the cytotoxicity of doranidazole by clonogenic assay, but the authors do not comment. Can they relate this to the extensive literature from the 1970s and 1980s on radiosensitisation and cytotoxicity of 2-nitroimidazoles in vitro? Might the treatment of cells embedded in agarose have limited drug exposure? (This procedure is not fully described in Methods). The concluding sentence on lines 139-141 is difficult to justify.

4. There is a disconnect between the radiosensitisation/cytotoxicity studies in Fig.1, in which hypoxia is <0.1% O₂, and the mechanistic studies in which hypoxia is 1% O₂. The term "hypoxia" is used interchangeably for these dramatically different conditions. 1% O₂ is not "hypoxia" as defined by radiation resistance, and is certainly too high an O₂ concentration to induce net bioreduction to the potentially cytotoxic hydroxylamine or amine metabolites. In contrast, significant redox cycling of doranidazole is highly unlikely to occur at <0.1% O₂. I.e. it would be helpful if the authors discussed the role of the well-known redox cycling of 2-nitroimidazoles under oxia or mild hypoxia in the induction of ferroptosis. I think their data show the mechanism of cytotoxicity is qualitatively the same under both conditions, and may be quite different to the mechanism of cytotoxicity under severe hypoxia (although this is not tested).

5. There is no discussion of the differential effects of doranidazole on TCA cycle intermediates. Why are some elevated and some depressed? Can this be related to the proposed ROS induction/ferroptosis mechanism?

6. The comparisons throughout of doranidazole with misonidazole and metronidazole, all at the same concentration, is of value in suggesting that bioreduction and redox cycling is involved, but the authors should explain this. Specifically, the lower one-electron reduction potential of metronidazole means that it has much lower potency than the 2-nitroimidazoles for any process dependent on its action as an oxidant (radiosensitisation, bioreductive cytotoxicity).

7. The Discussion goes beyond the data in its advocacy for human studies in GBM. This includes

lack of discussion of clinical studies of misonidazole as a radiosensitizer of GBM in the 1980s, none of which demonstrated utility. That is not to say that the early studies represent the last word – these did not identify patients with hypoxic tumours. But doranidazole was developed to be a less neurotoxic 2-nitroimidazole because of reduced BBB penetration, as acknowledged on lines 93-95. What are the implications of this for its use against GBM? It would be helpful to discuss critically the brain slice model in that context – it bypasses the likely BBB limitation.

Minor comments:

The Y axis label in Fig 1g,j would be normally expressed as “plating efficiency” (% or fraction).

21% O₂ is referred to throughout, but most or all of these experiments presumably used 5% CO₂ in the gas phase – although this is not specified. If so, the O₂ conc would have been 20%.

Fig 2c. What is the difference between the upper and lower image?

Becker cells? Explain what these are. An astrocytoma line?

Why are PI +ve cells extruded from the doranidazole-treated spheres? Comment?

The basal OCR values in the lower panels of Fig. 4a,b do not appear to correspond to the values in the upper panels.

ATP content in Fig. 4f is shown as % of control, but this precludes comparison of the 21% O₂ and 1% O₂ controls. This information would be useful, so I suggest normalising all to the oxic controls.

Line 270: “the adenylate change was reduced”. While this may be true, statistically, the change is trivial. How does this relate to the proposed mechanism and particularly the presumed decrease in OXPHOS?

The radiation dose rate(s) should be specified in Methods.

Please clarify whether drug exposures were continued during measurement of ATP (lines 493-496). What was the time lag between preparing cell suspensions and measurement of ATP with the CellTiter-Glo kit?

Two papers in Nature on 21 Oct report the oxidoreductase AIFM2 (renamed as FSP1) as a key suppressor of ferroptosis, independently of GPX4. The authors may want to take a look at the expression of both after doranidazole treatment in their transcriptomic analysis.

Signed: William R. Wilson

Reviewer #2 (Remarks to the Author):

Radiosensitization and toxicity studies on Doranidazole, a 2-nitroimidazole compound with less BBB permeability, was first published in 1995 (Ref.#15) and further characterized primarily by the same research groups and/or collaborators. The authors in this study continued to characterize toxicities of Doranidazole along with two other nitroimidazoles in glioma stem cell (GSC)-derived tumor in conjunction with IR and/or under normoxia and hypoxia conditions. The results using inhibitors and some signatures of (mitochondrial) oxidative stress suggest that Doranidazole exhibits toxicity in GSC-tumors at least in part through ferroptosis. Doranidazole also showed a trend of more toxicity in hypoxic tumor spheres or hypoxic conditions. Overall, the biochemical characterization in this work was extensive while they are more descriptive than mechanistic. The

findings in this study are seemingly new regarding the toxicity of Doranidazole with merit of development of therapeutic approaches; however, they will provide limited significance and scientific interests to the field. In addition, some of the results and approaches are not sufficient to support author's conclusion.

Major

1. Drug (Doranidazole) alone should have been tested in Fig. 1A rather than being performed separately in Fig. 1B.
2. Taken the results of the independent experiments in 1A and 1B together, 1) did IR contribute to additional 30 days of survival effects? If yes, provide the evidence for that. 2) How were the p-values in 1A and 1B obtained to support the author's conclusion of drug's effects positive in 1A but negative in 1B?
3. Figs. 1F, 1I graphs showing apparently no differences in surviving fraction between 0 and 6 mM Doranidazole under 0-6 Gy IR (but p values are <0.05) should be accurately noted and explained (lines 136-137).
4. The author's interpretation, lines 151-152 "Misonidazole but not metronidazole...", does not make sense from the results in Fig. S2A and 2B. Doranidazole should have been included in Fig. S2B and rephrase the statement appropriately.
5. The different susceptibility to Doranidazole cytotoxicity in Fig. 2F,G might be independent of hypoxic status of U251 and Becker spheres.
6. The results in Fig. 3A, by testing only a single concentration of each inhibitor, do not support author's interpretation "to a lesser extent by necrostatin-1" and "Z-VAD-FMK had no significant effect" (lines 192-193).
7. What are the effects of Doranidazole in early-stage or normal cell types?
8. Does Doranidazole deplete glutathione levels or inhibit GPX4? If not, does complex I or II inhibitor (e.g. rotenone) mimic Doranidazole?

Minor

- 1) Two staining panels in Fig. 2C are not labeled.
- 2) Western blots need positions of protein size markers.
- 3) Among only 46 citations, significant numbers are pretty old articles including more than decade-old review articles. Do these relatively older references represent the field of what authors have studied?
- 4) Doranidazole has primarily been studied by very limited numbers of research groups and collaborators, and the trend may not be changed (minor concerns of lower general interests in this particular (nice) work).

As the corresponding author, I am pleased to submit a revised version of our manuscript, "2-Nitroimidazoles induce mitochondrial stress and ferroptosis in glioma stem cells residing in a hypoxic niche".

We thank the editor and the reviewers for the time and effort invested in reviewing the data and for the insightful critiques which have helped improve our manuscript. According to the reviewers' constructive comments, we have added new data (presented in the main figures, supplementary figures, and figures for reviewers) and discussion points to address their concerns as detailed below and highlighted in yellow in the attached manuscript.

Response to the comments of Reviewer #1

1. *The first sentence of the Abstract is misleading. Radiosensitisation by nitroimidazoles is due to the parent drug, not their reduced metabolites. This error continues in the second sentence, and in the first paragraph of the Introduction. Second paragraph of the Introduction shows that the authors do understand the mechanisms behind the three reported properties of nitroimidazoles (radiosensitisation, hypoxia-selective cytotoxicity, hypoxia imaging), but the Abstract and first para need to be redrafted to make this clear.*

We thank the reviewer for pointing this out and the opportunity to correct any misleading parts. In the original draft, we have tried to summarize the properties of nitroimidazoles as much as possible within the word limit of the abstract, but we now see how the indicated paragraphs could be misleading.

The first paragraph in the abstract has been modified as follows: "Under hypoxic conditions, nitroimidazoles can replace oxygen as electron acceptors, thereby enhancing the effects of radiation on malignant cells. These compounds also accumulate in hypoxic cells, where they can act as cytotoxins. Whether these effects apply to cancer stem cells has not been sufficiently explored, however"

The introduction has also been modified to better detail and accentuate the difference between the three properties (radiosensitisation, hypoxia-selective

cytotoxicity, hypoxia imaging), p4 line 58.

2. The above notwithstanding, there is no clear evidence that any 2-nitroimidazole (including doranidazole) has therapeutic potential as a hypoxic cell cytotoxin (as distinct from activity as radiosensitizers). In that respect the mechanism of cytotoxicity, which is the focus of this study, is of limited interest, although it could be relevant to the well-studied neurotoxicity of 2-nitromidazoles (which is not mentioned by the authors).

We agree that there is no evidence that any 2-nitroimidazole has a therapeutic potential as a hypoxic cell cytotoxin as a single agent. We do however believe that the mechanism of cytotoxicity brings valuable information as to possible combination therapies and resistance mechanisms.

Specifically, our transcriptomics results show that doranidazole treatment induces upregulation of several REDOX-related genes, including aldehyde dehydrogenase Aldh3a1 (Fig. S3e). Aldh3a1 protects cells by detoxifying lipid peroxidation-derived reactive aldehydes such as 4-hydroxynonenal (4-HNE). Aldh3a1 can also release antioxidant nitric oxide (NO) from S-nitrosothiols, which can be formed after doranidazole administration. Studies from our lab have shown that upregulation of Aldh3a1 is a cause of resistance to ferroptosis induced by a different compound, xCT inhibitor sulfasalazine, and led to identifying aldehyde dehydrogenase inhibitors that show synthetic lethality with sulfasalazine (new references 48, 49: Okazaki et al, *Oncotarget* 2018, Otsuki et al, *Cancer Science* 2020).

We are now performing a similar screening for GSCs and doranidazole to identify combinations of synthetic lethality with maximal benefit in CNS malignancies.

We have extended our discussion to include this point (p18, line 378).

3. The authors report a surprising lack of radiosensitisation by doranidazole in vitro (Fig. 1F,I), even under severe hypoxia, despite the use of a very high concentration of the drug (6 mM). They claim statistically significant radiosensitisation, but I fail to see how they have determined such low p values. Anyway, the effect is biologically insignificant even if it is statistically significant. This is difficult to understand, as is the absence of hypoxic selectivity of the cytotoxicity of doranidazole by clonogenic assay, but the

authors do not comment. Can they relate this to the extensive literature from the 1970s and 1980s on radiosensitisation and cytotoxicity of 2-nitroimidazoles in vitro? Might the treatment of cells embedded in agarose have limited drug exposure? (This procedure is not fully described in Methods). The concluding sentence on lines 139-141 is difficult to justify.

We thank the reviewer for the valuable suggestion. As the reviewer suspected, the results partially reflected the limitations of the assay. Most of our GSC lines have a high migration ability and tend to aggregate, which makes traditional colony formation assays difficult and has required the use of several layers of agarose in our previous work. However, agarose embedding does indeed limit drug exposure for certain compounds. This is the reason a higher concentration (6 mM) was used in the initial studies.

We have now re-evaluated and optimized the conditions for the colony formation assay for the GSCs used in this study and succeeded in performing the assay without the use of agarose. Furthermore, according to the reviewer's suggestion, we carefully reviewed the literature from the 1970s and 1980s and found that the irradiation doses used were usually higher than those in our study. We have now also increased the radiation dose accordingly.

The new data shows a more pronounced radiosensitization effect and hypoxia selectivity for 3 mM doranidazole. The results are presented in Fig. 1d-g and the methods section has also been modified.

4. There is a disconnect between the radiosensitisation/cytotoxicity studies in Fig.1, in which hypoxia is <0.1% O₂, and the mechanistic studies in which hypoxia is 1% O₂. The term "hypoxia" is used interchangeably for these dramatically different conditions. 1% O₂ is not "hypoxia" as defined by radiation resistance, and is certainly too high an O₂ concentration to induce net bioreduction to the potentially cytotoxic hydroxylamine or amine metabolites. In contrast, significant redox cycling of doranidazole is highly unlikely to occur at <0.1% O₂.

We thank the reviewer for the opportunity to further clarify this matter. We have replaced "hypoxia" with the O₂ concentration wherever possible.

The reason most mechanistic studies were performed in 1% O₂ are reports indicating that mild hypoxia increases stem cell fraction and proliferation and findings from clinical samples which detected GSCs mainly in regions of

intermediate oxygenation. This explanation has also been added to the manuscript on page 7, line 138.

I.e. it would be helpful if the authors discussed the role of the well-known redox cycling of 2-nitroimidazoles under oxia or mild hypoxia in the induction of ferroptosis. I think their data show the mechanism of cytotoxicity is qualitatively the same under both conditions, and may be quite different to the mechanism of cytotoxicity under severe hypoxia (although this is not tested).

Thank you for the very valuable feedback. As the reviewer points out, the transcriptomics results, extracellular flux analysis and ROS measurements suggest that the mechanism of cytotoxicity is qualitatively the same for normoxia and mild hypoxia. This point has also been added to the discussion (p17 line 370).

For a direct comparison between the degree of cytotoxicity in the three conditions, we have performed additional experiments and added the results for a 24-hour treatment. We also suspect that cytotoxicity in 0.1% O₂ might include other mechanisms and hope to answer this question in our future work.

5. There is no discussion of the differential effects of doranidazole on TCA cycle intermediates. Why are some elevated and some depressed? Can this be related to the proposed ROS induction/ferroptosis mechanism?

As the reviewer points out, most of the changes in TCA cycle metabolites are related to the ROS induction and ferroptosis mechanism. Taken together with the elevated levels of the NADH/NAD⁺ ratio, which inhibit dehydrogenases in the TCA cycle, the increase in succinate and decrease in malate and fumarate are consistent with a decrease in activity of the ETC, particularly mitochondrial complex II. We have added this point to the manuscript on page 13, line 278.

Furthermore, the decrease in α -ketoglutarate is consistent with the decrease in glutamine and glutamate, suggesting a decrease in glutaminolysis. The decrease in glutamine and glutamate levels could also contribute to increased ROS and ferroptosis through a decrease in GSH. Further dissection of the specific contribution of each ROS-producing pathway after doranidazole treatment could reveal new combination therapies.

6. The comparisons throughout of doranidazole with misonidazole and metronidazole, all that the same concentration, is of value in suggesting that bioreduction and redox cycling is involved, but the authors should explain this. Specifically, the lower one-electron reduction potential of metronidazole means that it has much lower potency than the 2-nitroimidazoles for any process dependent on its action as an oxidant (radiosensitisation, bioreductive cytotoxicity).

We thank the reviewer for this suggestion. We have now performed additional experiments with equimolar concentrations of etanidazole and nimorazole (new fig2f) and found that their effect (cell death induction by etanidazole, but not nimorazole) was also consistent with their reported one-electron reduction potential. We have also added this point to the discussion (p18, line 393).

7. The Discussion goes beyond the data in its advocacy for human studies in GBM. This includes lack of discussion of clinical studies of misonidazole as a radiosensitiser of GBM in the 1980s, none of which demonstrated utility. That is not to say that the early studies represent the last word – these did not identify patients with hypoxic tumours. But doranidazole was developed to be a less neurotoxic 2-nitroimidazole because of reduced BBB penetration, as acknowledged on lines 93-95. What are the implications of this for its use against GBM? It would be helpful to discuss critically the brain slice model in that context – it bypasses the likely BBB limitation.

Our results suggest that doranidazole would only be effective against very specific regions in a tumor: regions that are hypoxic, have lost BBB function and are enriched in glioma stem cells. We did not intend to advocate for human studies, but to highlight that intrinsic GSCs resistance to nitroimidazoles is not likely to have been a major factor in the failure of the clinical studies and to stress, as the reviewer mentions, the importance of identifying highly hypoxic tumors.

We have rewritten the paragraph to clarify the point and included the references for clinical studies with misonidazole. The use of the brain slice model to assess drugs that do not pass the BBB has been the object of our previous work (ref.35). The fact that the slice model bypasses the BBB

limitation has thus been added to the main text with the corresponding reference (p15, line 333).

Minor comments:

The Y axis label in Fig 1g,j would be normally expressed as “plating efficiency” (% or fraction).

We have modified the label accordingly.

21% O₂ is referred to throughout, but most or all of these experiments presumably used 5% CO₂ in the gas phase – although this is not specified. If so, the O₂ conc would have been 20%.

We thank the reviewer for this important point. The figures and manuscript have been revised to specify the culture conditions and to indicate the actual, measured O₂ concentrations, which were indeed 20%.

Fig 2c. What is the difference between the upper and lower image?

The images showed spheres of different sizes. For clarity, we have removed the lower panel.

Becker cells? Explain what these are. An astrocytoma line?

Becker is a human astrocytoma grade III-IV cell line. We have added the explanation to the methods section (p19 line 424).

Why are PI +ve cells extruded from the doranidazole-treated spheres? Comment?

The extrusion of the dead cells is a very important finding that we are actively analyzing. So far, we found that a) it also occurs in non-transformed Ink4a/Arf KO NSC spheres b) the effect is not seen in doranidazole-treated human GSC spheres c) drugs which induce cell-death uniformly within the sphere do not have a similar effect. These results suggest that the phenomenon is not Ras-dependent and that it might be related to cell competition in the context of metabolic catastrophe, possibly in the absence of *p16/p19*. We will continue to investigate the mechanism and hope to report the results in our future work.

The basal OCR values in the lower panels of Fig. 4a,b do not appear to

correspond to the values in the upper panels.

The apparent discrepancy is probably due to two reasons:

1. the upper panels are the graph from one representative experiment, while the lower panels represent the quantifications (means \pm SD) from three independent experiments,

2. the upper panels show the serial OCR values for all time points, while the basal OCR values are subtraction values (last rate measurement before first injection – non-mitochondrial respiration rate).

It is therefore difficult to compare values between graphs. We have modified the figure legend to further clarify these points.

The values used for the lower panels in fig.4a,b are presented below.

20% O ₂	PBS			Doranidazole 1 m M			Doranidazole 3 m M		
	n=1	n=2	n=3	n=1	n=2	n=3	n=1	n=2	n=3
Basal	254.8	193.00	233.6	184.5	142.63	159.8	129.1	125.00	132.5
Proton Leak	64.9	69.33	52.9	45.1	46.92	40.2	37.9	35.90	42.3
Maximal Respiration	345.8	271.65	344.7	213.7	154.34	204.6	101.1	76.58	115.2
Non Mitochondrial Oxygen Consumption	32.5	38.01	53.3	32.3	39.02	51.6	48.5	42.42	39.5
ATP Production	189.9	123.68	180.8	139.4	95.71	119.6	91.1	89.10	90.2
Spare Respiratory Capacity (%)	135.0%	143.61%	147.7%	116.9%	108.59%	127.0%	77.4%	62.34%	87.6%
1% O ₂	PBS			Doranidazole 1 m M			Doranidazole 3 m M		
	n=1	n=2	n=3	n=1	n=2	n=3	n=1	n=2	n=3
Basal	155.3	126.8	165.1	121.8	100.3	120.1	78.0	39.7	39.2
Proton Leak	23.9	21.8	22.5	20.6	18.7	20.2	35.5	23.1	10.8
Maximal Respiration	294.7	252.0	248.6	209.9	157.7	170.0	57.4	26.9	32.2
Non Mitochondrial Oxygen Consumption	45.3	35.7	43.8	38.2	21.1	27.7	21.1	14.0	53.3
ATP Production	131.4	105.0	142.6	101.2	81.6	99.9	42.5	16.6	28.4
Spare Respiratory Capacity (%)	189.7%	197.9%	149.4%	174.4%	154.5%	145.4%	71.3%	67.1%	84.7%

ATP content in Fig. 4f is shown as % of control, but this precludes comparison of the 21% O₂ and 1% O₂ controls. This information would be useful, so I suggest normalising all to the oxic controls.

The figure has been revised accordingly and is now presented as the new fig.4f. The figure presents data from three independent experiments with all groups evaluated in parallel to allow direct multiple comparisons. Interestingly, we found that exposure to 1% O₂ for 12 h did not significantly change ATP levels, consistent with reports from hypoxia tolerant GBM cell lines (Turcotte et al, *Brit J of Cancer* 2002).

In accordance with the change in figure and to the reviewers' overall suggestions regarding statistical significance, we have also modified the

main text to highlight only differences that were both robust and statistically significant.

Line 270: “the adenylate change was reduced”. While this may be true, statistically, the change is trivial. How does this relate to the proposed mechanism and particularly the presumed decrease in OXPHOS?

Our CE-MS metabolome data also showed that after a 24-hour treatment with doranidazole, the impairment of the mitochondrial respiration resulted in significantly decreased ATP and increased AMP levels (new fig.5c), and a subsequent significant but modest drop of the adenylate charge. These changes were not seen after metronidazole treatment. Fig. 5b-c, fig. S5 and the result section have been modified to present these changes in more detail and in a visually consistent manner.

In addition, to address the reviewer’s concern, we reexamined whether any compensation mechanisms might have contributed to prevent a more robust decrease of the adenylate charge. Interrogation of the microarray samples acquired in parallel with the metabolome samples showed a modest, but significant upregulation of glycolytic enzymes *Hk1* (fold change for doranidazole 3mM vs PBS : 1.9, $p=0.0005$) and *Pfkfb3* (fold change for doranidazole 3mM vs PBS : 1.39, $p=0.0218$) after 24h, suggesting a possible compensation by an increase in glycolysis activity.

Given that compensation mechanisms could play a crucial role in resistance to nitroimidazoles, we will address this point in detail in our future work.

The radiation dose rate(s) should be specified in Methods.

We have added the radiation dose rates (p20 line 446 and p23 line 521).

Please clarify whether drug exposures were continued during measurement of ATP (lines 493-496). What was the time lag between preparing cell suspensions and measurement of ATP with the CellTiter-Glo kit?

Drug exposure was not continued during ATP measurements and there was a time lag of 30 min. We have added the detailed information to the methods section (p24, line 547).

Two papers in Nature on 21 Oct report the oxidoreductase AIFM2 (renamed

as *FSP1*) as a key suppressor of ferroptosis, independently of *GPX4*. The authors may want to take a look at the expression of both after doranidazole treatment in their transcriptomic analysis.

As recommended by the reviewer, we have interrogated the transcriptome analysis, but found no significant difference in the expression of *AIFM2* in our cells, as shown below.

Reviewer #2 (Remarks to the Author):

Major

1. Drug (Doranidazole) alone should have been tested in Fig. 1A rather than being performed separately in Fig. 1B.

2. Taken the results of the independent experiments in 1A and 1B together, 1) did IR contribute to additional 30 days of survival effects? If yes, provide the evidence for that. 2) How were the *p*-values in 1A and 1B obtained to support the author's conclusion of drug's effects positive in 1A but negative in 1B?

To address these points, we have performed a four-armed study to include all groups in a single experiment. To ensure quality control of our cells during the hours necessary for all implantations (which was the main concern and the reason the initial experiments were performed separately), we periodically refreshed cell suspensions and confirmed cell viability. As before, animals have been randomized into the four groups.

The new survival curves were analyzed with the log-rank test. The results show that drug alone does not prolong survival (median survival control vs

drug : 28.5 vs 29 days), while drug administration before IR prolongs survival compared to IR (median survival IR vs IR + drug: 48 vs 53 days). In the four-armed study, we did indeed find that IR alone increased survival (median survival control vs IR: 28.5 vs 48 days). The effect of IR alone on GSCs was similar to the one we have previously reported (Osuka et al, *Stem Cells* 2013) and therefore not highlighted further in the present study. These results are now presented as the new Fig.1a.

3. Figs. 1F, 1I graphs showing apparently no differences in surviving fraction between 0 and 6 mM Doranidazole under 0-6 Gy IR (but p values are <0.05) should be accurately noted and explained (lines 136-137).

Our original results were obtained from an assay using agarose embedding to prevent cell movement and aggregation. However, agarose limited drug exposure and made interpretation of the results difficult. We have now performed classical colony formation assays and in accordance with historical studies, have also added higher doses of radiation. The *P* values for the comparison of the curves have been replaced by indication of significance for the comparison between individual points. The new results are presented as Fig. 1d-g. We have also added the detailed explanation of the *P* value to the figure legend.

4. The author's interpretation, lines 151-152 "Misonidazole but not metronidazole...", does not make sense from the results in Fig. S2A and 2B. Doranidazole should have been included in Fig. S2B and rephrase the statement appropriately.

According to the reviewer's suggestion, we have included doranidazole in fig. S2b and directly compared the effects of the 3 drugs. We also performed two-tailed ANOVA followed by Tukey's post-hoc test. While the results for doranidazole vs PBS remained significant in the new analysis, the differences for metronidazole and misonidazole failed to retain statistical significance in a multiple comparison setting. We have therefore amended the manuscript to "Equimolar concentrations of metronidazole and misonidazole did not affect cell cycle distribution" (p8 lines 151).

5. The different susceptibility to Doranidazole cytotoxicity in Fig. 2F, G might be independent of hypoxic status of U251 and Becker spheres.

Thank you for the valuable feedback and the opportunity to clarify this point. We have performed additional analyses and found that, in monolayer culture, doranidazole induces cell death in both U251 and Becker in severe hypoxia (0.1% O₂, EF5 binding 30%), but not in mild hypoxia (1% O₂, EF5 binding <10%). Taken together with the staining patterns of EF5 in TSH, U251 and Becker cells, these results support a correlation between cell death and the level of hypoxia present in the spheres. The results have been added as the new figure in S2i, j.

6. The results in Fig. 3A, by testing only a single concentration of each inhibitor, do not support author's interpretation "to a lesser extent by necrostatin-1" and "Z-VAD-FMK had no significant effect" (lines 192-193). According to the reviewer's suggestion, we have added our results from testing multiple concentrations in the new fig. S3a.

7. What are the effects of Doranidazole in early-stage or normal cell types?

To address this question, we made use of our NSC, which, upon serum addition, differentiate along astrocytic lines (Fig. S1b). We found that doranidazole only induced significant cell death in conditions of 0.1% O₂, levels to which normal cells are not exposed in the brain in physiological conditions. These results have now been added as fig. S2g.

8. Does Doranidazole deplete glutathione levels or inhibit GPX4? If not, does complex I or II inhibitor (e.g. rotenone) mimic Doranidazole?

As the reviewer points out, the effect of doranidazole on glutathione is a very important issue. Depletion of thiols has indeed been reported and proposed as one of the mechanisms of nitroimidazole cytotoxicity, although this effect might be more accentuated in vitro than in vivo (Wardman et al, *Brit J Radiol* 2018).

To address the reviewer's question, we have now performed several additional analyses.

In experiments with GSC spheres, we found that addition of exogenous GSH had a slight tendency to rescue toxicity in spheres, but the rescue was less than that of ferrostatin or N-acetylcysteine.

LC-MS metabolome analysis showed that doranidazole slightly decreased reduced glutathione (GSH) and total glutathione (GSH+2xGSSG) levels in

normoxic conditions. (Units for metabolite levels: $\mu\text{mol/g}$ protein)

However, further interrogation of LC-MS results revealed a significant increase in antioxidant hypotaurine in doranidazole-treated GSCs, on the background of dramatically decreased serine levels. This result is consistent with our previous findings in the same GBM model, which showed that a decrease in GSH induces a compensatory increase in alternate antioxidant pathways, including hypotaurine (Shiota et al, *Nat Commun*, 2018, ref 51).

The upregulation of antioxidant pathways could play a crucial role in resistance to nitroimidazoles and we therefore plan to address this point in detail in our future work.

However, these results strongly suggest, as the reviewer suspected, that changes induced by doranidazole are indeed multifactorial. Therefore, complex I and II inhibitors alone did not have a similar biological effect, as shown below. Regarding complex II, inhibition of ROS production at the flavin site was less amenable to testing due to the poor intracellular

penetration of succinate analogues, but inhibition at the ubiquinone-binding site did not have marked cytotoxic effects. Further dissection of the contribution of each ROS-producing pathway after doranidazole treatment could reveal new combination therapies. We are now actively investigating this issue and hope to address it in our future work.

Complex I inhibitors

Complex II inhibitors

Minor

1) *Two staining panels in Fig. 2C are not labeled.*

The images showed spheres of different sizes. For clarity, we have removed the lower panel.

2) *Western blots need positions of protein size markers.*

We have added the markers.

3) Among only 46 citations, significant numbers are pretty old articles including more than decade-old review articles. Do these relatively older references represent the field of what authors have studied?

We would like to thank the reviewer for the careful assessment of the reference section. A substantial part of the investigations related to the use of nitroimidazoles in cancer has been reported more than one decade ago and the results of those investigations are considered valid to the present day. Whenever possible, we have tried to include the articles which originally presented the information quoted. For review articles, we have chosen the

ones which provide an original point of view that was later quoted by newer reviews or had the most detailed discussion regarding the quoted point.

REVIEWERS' COMMENTS:

Reviewer #1 (Remarks to the Author):

The data have been strengthened by additions in the revised ms. Particularly Fig. 1d-g and the new Fig. 2f and 5c all of which are instructive (although I have noted a minor issue with Fig. 1f,g below). In addition, some of the limitations of the study (such as bypassing of the BBB limitation in the slice model) are now clearly acknowledged. My remaining concerns relate mainly to interpretation of the data.

As I commented previously, there is little or no evidence that 2-nitroimidazoles can act as cytotoxins in tumours at pharmacologically meaningful doses. One of my main comments on the original draft was that the significance of the study is therefore likely to relate to the oxidative cytotoxicity (particularly neurotoxicity) of these compounds rather than their antitumour activity. The authors state on lines 72-75 that 2-nitroimidazoles cause radiation-independent cytotoxic effects in hypoxic cells, citing two papers from the 1980s – one of which actually relates to radiosensitisation rather than cytotoxicity. The other (ref 8) used heroic doses of the compounds in mice.

In the rebuttal letter, it is suggested that tumour cell cytotoxicity may be important in combination settings, which is an old idea that was explored in the early 1980s. 2-nitroimidazoles have not found use clinically in any combination setting other than radiotherapy (where they are acting as electron-affinic radiosensitisers, not cytotoxins, and even that context they are not in widespread use). However, the authors appear to be committed to pursuing combination interactions in the future, so I suppose they are justified in making that argument.

The above concern does not take away from the importance of the authors' observations about induction of mitochondrial ROS and ferroptosis. I just think that the significance relates to normal tissue toxicities rather than antitumour activity.

Specific comments:

In the Abstract, line 40. Notwithstanding the above, I suggest that "...where they can act as cytotoxins or imaging agents" would be more consistent with this manuscript.

The hypothesis that "the limited therapeutic effect of 2-nitroimidazoles might be due to intrinsic resistance of cancer stem cells to this class of compounds" has little to commend it. In the context, the authors appear to be talking about radiosensitisation by 2-nitroimidazoles (?) which is an essentially universal phenomenon in all hypoxic cells as the mechanism is a very simple chemical reaction (oxidation of radiation-induced DNA free radicals) which is not modified by anything with the exception of high concentrations of thiols (GSH, CySH). Perhaps the authors are referring to cytotoxicity in the absence of radiation, but if so this should be made clear.

In Fig. 1 f,g the lines drawn to indicate the statistical test imply that it is the model-fitted values that were used. Obviously the experimental values must be used, so the lines should be redrawn.

Line 60: "The electrophilic properties of nitroimidazoles allow them to replace O₂ in fixating the radiation-induced production of reactive oxygen species (ROS)". This is wrong. Nitroimidazoles oxidise radiation-induced DNA radicals, they do not "fixate" ROS. The radiation induced ROS are the hydroxyl radical (which arises from radiolysis of water, not O₂) and superoxide. Nitroimidazoles do not replace O₂ in the generation of superoxide.

Line 65. "compounds are immediately reoxygenated and removed from cells". Reoxidised, not reoxygenated. (Oxygen is not incorporated). And, delete "removed from cells". The parent 2-nitroimidazole is regenerated, not removed.

I suggest a more cautious interpretation on lines 134-5 as to the mechanism for in vivo activity in Fig. 1a, which is interpreted as radiosensitisation based on Fig. 1d-e. Although radiosensitisation in vitro was statistically significant, and cytotoxicity was not, the magnitude of both effects was tiny and whether the effects are statistically significant reflects the variance in the assays as well as the magnitude of effect. Although I have argued above that 2-nitroimidazoles are not effective hypoxia-selective cytotoxins in vivo, it is not possible to establish clearly from the present data what mechanism(s) are operative in Fig. 1a. (Greater activity with radiation is expected with hypoxia-selective cytotoxins as well as radiosensitisers).

Line 173-178 is misleading: "Serum-treated NSCs, which differentiate along the astrocytic line (Supplementary Fig. 1b) and are expected to reside in the normal brain at physiological oxygen concentrations of 0.5% to 7%²⁵, did not undergo significant cell death after exposure to doranidazole in the presence of 20% or 1% O₂ (Supplementary Fig. 2g)." In fact their data show no clear difference between GSH-H and serum-treated NSC in this respect. (GSC-H cells also did not show significant cell killing by doranidazole under 20% or 1% O₂ in Supplementary Fig. 2f).

Reviewer #2 (Remarks to the Author):

The authors successfully addressed all concerns by including new and convincing experimental data.

Minor

Although EF5 was introduced earlier (line 69), EF binding (lines 138-140) may need a short note of hypoxia measure as the Methods section come after discussion and might be neglected.

As the corresponding author, I am pleased to submit a revised version of our manuscript, "2-Nitroimidazoles induce mitochondrial stress and ferroptosis in glioma stem cells residing in a hypoxic niche".

We thank the editor and the reviewers for the insightful suggestions that have helped improve our manuscript. We have now better explained the parts that were considered potentially misleading and added discussion points to address the concerns as detailed below and highlighted in yellow in the attached manuscript.

Response to the comments of Reviewer #1

As I commented previously, there is little or no evidence that 2-nitroimidazoles can act as cytotoxins in tumours at pharmacologically meaningful doses. One of my main comments on the original draft was that the significance of the study is therefore likely to relate to the oxidative cytotoxicity (particularly neurotoxicity) of these compounds rather than their antitumour activity. The authors state on lines 72-75 that 2-nitroimidazoles cause radiation-independent cytotoxic effects in hypoxic cells, citing two papers from the 1980s – one of which actually relates to radiosensitisation rather than cytotoxicity. The other (ref 8) used heroic doses of the compounds in mice.

We agree with the reviewer that "there is little or no evidence that 2-nitroimidazoles can act as cytotoxins in tumours at pharmacologically meaningful doses". Lines 72-75 were meant to acknowledge existing reports of radiation-independent cytotoxic effects. For clarity, the paragraph has been rephrased (line 58 in revised manuscript). We have also added a paragraph discussing the significance of the findings in the context of oxidative cytotoxicity towards normal tissues in the discussion.

In the rebuttal letter, it is suggested that tumour cell cytotoxicity may be important in combination settings, which is an old idea that was explored in the early 1980s. 2-nitroimidazoles have not found use clinically in any combination setting other than radiotherapy (where they are acting as electron-affinic radiosensitisers, not cytotoxins, and even that context they

are not in widespread use). However, the authors appear to be committed to pursuing combination interactions in the future, so I suppose they are justified in making that argument.

As the reviewer points out, combination therapy has been explored in the 1980s and indeed, no combination setting other than radiotherapy has reached clinical use. However, we believe that our new data can lead to testing combinations that have not been tested before and that it warrants further exploration of this possibility in preclinical models.

The above concern does not take away from the importance of the authors' observations about induction of mitochondrial ROS and ferroptosis. I just think that the significance relates to normal tissue toxicities rather than antitumour activity.

This is an important point. The significance of the findings as related to normal tissue toxicity is now discussed on lines 398-407.

Specific comments:

In the Abstract, line 40. Notwithstanding the above, I suggest that "...where they can act as cytotoxins or imaging agents" would be more consistent with this manuscript.

We have amended the abstract as suggested.

The hypothesis that "the limited therapeutic effect of 2-nitroimidazoles might be due to intrinsic resistance of cancer stem cells to this class of compounds" has little to commend it. In the context, the authors appear to be talking about radiosensitisation by 2-nitroimidazoles (?) which is an essentially universal phenomenon in all hypoxic cells as the mechanism is a very simple chemical reaction (oxidation of radiation-induced DNA free radicals) which is not modified by anything with the exception of high concentrations of thiols (GSH, CySH). Perhaps the authors are referring to cytotoxicity in the absence of radiation, but if so this should be made clear.

This hypothesis was based mainly on two points:

1. Cancer stem cells have been reported to possess a high ability to repair DNA-damage induced by radiation. This ability could lead to an increased threshold for the degree of radiosensitization necessary to achieve clinical

benefit.

2. Cancer stem cells have been reported to have increased levels of multi-drug resistance-associated protein (MRP1). MRP1 is used to export glutathione-conjugates of various substrates out of the cells (Ishikawa et al, Cytotechnology, 1998). Levels of *MRP1* have been shown to be inversely correlated with accumulation of FMISO (as the glutathione conjugate of reduced FMISO, amino-FMISO-GS) in hypoxic cells (Masaki et al, Ann Nucl Med 2017). Increased levels of MRP1 in cancer stem cells could thus lead to resistance through a higher efflux of glutathione-conjugated nitroimidazole derivatives.

In Fig. 1 f,g the lines drawn to indicate the statistical test imply that it is the model-fitted values that were used. Obviously the experimental values must be used, so the lines should be redrawn.

The figure now presents the lines for the experimental values, as does Fig.S1e.

Line 60: “The electrophilic properties of nitroimidazoles allow them to replace O₂ in fixating the radiation-induced production of reactive oxygen species (ROS)”. This is wrong. Nitroimidazoles oxidise radiation-induced DNA radicals, they do not “fixate” ROS. The radiation induced ROS are the hydroxyl radical (which arises from radiolysis of water, not O₂) and superoxide. Nitroimidazoles do not replace O₂ in the generation of superoxide.

The manuscript has been corrected to state “The electrophilic properties of nitroimidazoles allow them to oxidize the radiation-induced DNA radicals⁴⁻⁶, thereby enhancing the effects of radiation.”

Line 65. “compounds are immediately reoxygenated and removed from cells”. Reoxidised, not reoxygenated. (Oxygen is not incorporated). And, delete “removed from cells”. The parent 2-nitroimidazole is regenerated, not removed.

The manuscript has been corrected to state “In the presence of O₂, the reduced metabolites of these compounds are immediately reoxidized, resulting in redox cycling of the drugs.”

I suggest a more cautious interpretation on lines 134-5 as to the mechanism for in vivo activity in Fig. 1a, which is interpreted as radiosensitisation based on Fig. 1d-e. Although radiosensitisation in vitro was statistically significant, and cytotoxicity was not, the magnitude of both effects was tiny and whether the effects are statistically significant reflects the variance in the assays as well as the magnitude of effect. Although I have argued above that 2-nitroimidazoles are not effective hypoxia-selective cytotoxins in vivo, it is not possible to establish clearly from the present data what mechanism(s) are operative in Fig. 1a. (Greater activity with radiation is expected with hypoxia-selective cytotoxins as well as radiosensitisers).

We share the reviewers' opinion that great caution is needed in interpreting the extent of the cytotoxic effect and completely agree that it is not possible to clearly establish the exact contribution of radiosensitization vs cytotoxicity. However, in the manuscript we merely state that radiosensitization is observed in our GSC-based tumors, which we have confirmed in repeated animal experiments.

Line 173-178 is misleading: "Serum-treated NSCs, which differentiate along the astrocytic line (Supplementary Fig. 1b) and are expected to reside in the normal brain at physiological oxygen concentrations of 0.5% to 7%²⁵, did not undergo significant cell death after exposure to doranidazole in the presence of 20% or 1% O₂ (Supplementary Fig. 2g)." In fact their data show no clear difference between GSH-H and serum-treated NSC in this respect. (GSC-H cells also did not show significant cell killing by doranidazole under 20% or 1% O₂ in Supplementary Fig. 2f).

In light of the reviewers' comment and related comments on normal tissue toxicity, "and are expected to reside in the normal brain at physiological oxygen concentrations of 0.5% to 7%" has been removed from lines 173-178 and the paragraph now states "Serum-treated NSCs, which differentiate along the astrocytic line (Supplementary Fig. 1b), underwent significant cell death after exposure to doranidazole in the presence of 0.1% O₂, but not in the presence of 20% or 1% O₂ (Supplementary Fig. 2g)."

A new paragraph on normal tissue toxicity has been added to the discussion to address these findings (lines 398-407).

Reviewer #2 (Remarks to the Author):

Minor

Although EF5 was introduced earlier (line 69), EF binding (lines 138-140) may need a short note of hypoxia measure as the Methods section come after discussion and might be neglected.

Lines 138-140 (new lines 125-126) have been changed to specify “In addition to regions of severe hypoxia or anoxia (<0.1% O₂; binding of hypoxia-marker EF5, 30–100%)²⁴, which are primary targets of radiosensitization...”.